# Make Haste Slowly: A Theory of Emergent Structured Mixed Selectivity in Feature Learning ReLU Networks

**Devon Jarvis**[1,2*]**, Richard Klein**[1,2]**, Benjamin Rosman**[1,2,4] **& Andrew M. Saxe**[3,4]

[1]School of Computer Science and Applied Mathematics, University of the Witwatersrand
[2]Machine Intelligence and Neural Discovery Institute, University of the Witwatersrand
[3]Gatsby Computational Neuroscience Unit & Sainsbury Wellcome Centre, UCL
[4]CIFAR Azrieli Global Scholar, CIFAR
`{devon.jarvis,richard.klein,benjamin.rosman1}@wits.ac.za`
`a.saxe@ucl.ac.uk`

## Abstract

In spite of finite dimension ReLU neural networks being a consistent factor behind recent deep learning successes, a theory of feature learning in these models remains elusive. Currently, insightful theories still rely on assumptions including the linearity of the network computations, unstructured input data and architectural constraints such as infinite width or a single hidden layer. To begin to address this gap we establish an equivalence between ReLU networks and Gated Deep Linear Networks, and use their greater tractability to derive dynamics of learning. We then consider multiple variants of a core task reminiscent of multi-task learning or contextual control which requires both feature learning and nonlinearity. We make explicit that, for these tasks, the ReLU networks possess an inductive bias towards latent representations which are not strictly modular or disentangled but are still highly structured and reusable between contexts. This effect is amplified with the addition of more contexts and hidden layers. Thus, we take a step towards a theory of feature learning in finite ReLU networks and shed light on how structured mixed-selective latent representations can emerge due to a bias for node-reuse and learning speed.

## 1 Introduction

When contrast against the rapid empirical progress of neural networks (Mnih et al., 2015; Amodei et al., 2016; Andreas et al., 2016; He et al., 2016; Silver et al., 2017; Vaswani et al., 2017; Baevski et al., 2020; Reid et al., 2024) it can appear that theoretical understanding of these models is not keeping pace. However, great theoretical progress *has* been made with a number of new paradigms being proposed and investigated (Jacot et al., 2018; Goldt et al., 2020; Saxe et al., 2022; Li & Sompolinsky, 2022). While these paradigms illuminate many aspects of deep network behaviour, they typically simplify feature learning – a key ingredient thought to underlie the success of deep networks (Krizhevsky et al., 2017; Jing & Tian, 2020). For instance, while statistical physics provides a theoretical paradigm with clear insight for unstructured data from Gaussian distributions (Saad & Solla, 1995; Riegler & Biehl, 1995; Goldt et al., 2019; Advani et al., 2020), treating more intricately structured domains remains challenging. Consequently, these theories can certainly explain phenomena observed in practice (Ramasesh et al., 2020; Lee et al., 2022; 2024) but do not explicitly capture dynamics with richer data structure (Goldt et al., 2020). Similarly, the Neural Tangent Kernel (NTK) (Jacot et al., 2018; Lee et al., 2019) offers exact learning dynamics for network architectures trained in the "lazy regime" of infinite hidden layer width or large initial weights (Chizat et al., 2019; Alemohammad et al., 2021). Yet, lazy networks tend to generalize worse than those in the "feature learning regime" (Bietti & Mairal, 2019; Geiger et al., 2020) of small initial weights and finite width. Conversely, deep linear networks (Saxe et al., 2014; 2019) offer a theory of feature learning and have successfully characterised continual learning (Braun et al., 2022) and systematic generalisation (Jarvis et al., 2023). Yet, these models

---

[*]Corresponding author

do not incorporate nonlinear activation functions which are necessary for the network to perform real world tasks ranging in complexity from the simple XoR (Minsky & Papert, 1969) to Context Generalization (Sodhani et al., 2021; Dahl et al., 2011; Beukman et al., 2024). Other useful approaches consider a static analysis of stable points (Seung et al., 1992; Zdeborová & Krzakala, 2016) or require infinite input dimension with infinite data (known as the thermodynamic limit) (Mignacco et al., 2020; Goldt et al., 2020; Li & Sompolinsky, 2022). Hence an explicit theory of feature learning in ReLU networks (Fukushima, 1969; Nair & Hinton, 2010) with finite width has not yet been presented, in spite of their consistent use in the empirical successes of deep learning (Nwankpa et al., 2018).

In this work we take a step towards a theory of feature learning in finite ReLU networks. We do this by drawing an equivalence between ReLU networks and Gated Deep Linear Networks (GDLN) (Saxe et al., 2022) in Section 3. As shown in Figure 1, GDLNs provide a class of network architectures which perform nonlinear computations on their inputs using a composition of linear modules and are amenable to a full characterisation of their training dynamics. We show that for a ReLU network it is possible to create an equivalent GDLN which has the same output as the ReLU network at all points in time. We term this the "*Rectified Linear Network (ReLN)*" and obtain the full training dynamics of both networks trained on an adapted XoR task. Section 4 then applies this technique to a significantly more realistic setting reminiscent of multi-task learning or contextual control (Zamir et al., 2018). We provide full training dynamics for the ReLU network in this setting via the ReLN and prove that in this case the equivalence is unique. In this task, we find that ReLU networks present an inductive bias towards structured mixed-selective latent representations which are reused across contexts. Section 5 and Section 6 then consider the effect of adding contexts and multiple hidden layers re-

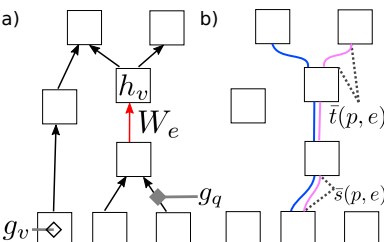

Figure 1: GDLN Formalism and notation. a) The GDLN applies gating variables to nodes ($g_v$) and edges ($g_q$) in an otherwise linear network. b) The gradient for an edge (using the red edge in (a) for example) can be written in terms of paths through that edge (colored lines). Each path is broken into the component preceding ($\bar{s}(p, e)$) and following ($\bar{t}(p, e)$) the edge.

spectively. We find that the bias towards structured mixed selectivity and node reuse is amplified by these changes, indicating that this may be a consistent emergent strategy for tasks of increasing difficulty. Thus, we summarize our main contributions as follows:

- We introduce Rectified Linear Networks (ReLNs) as the subset of GDLNs which each imitate a ReLU network. Through this we obtain the full training dynamics for finite, feature learning ReLU networks.

- We demonstrate that, in a contextual task, ReLU networks have an implicit bias towards structured mixed selectivity even during feature learning as this minimizes the loss the fastest.

- Finally, we demonstrate that the bias towards node reuse is exasperated with the addition of more contexts and hidden layers on this task.

Thus, while performing "slow" feature learning mixed-selective functional modules emerge in ReLU networks which exploit node-reuse for learning speed (or "haste").

## 2 BACKGROUND

We briefly review the GDLN paradigm here as our work builds directly from it. However, the paradigm itself builds on the linear neural network theory (Saxe et al., 2014; 2019). Thus, for readers being introduced to linear neural network theory from this work we provide a more thorough review in Appendix A. The GDLN notation and formalism is also depicted in Figure 1. GDLNs are a class of neural network which compute nonlinear functions of their inputs through the composition of linear network modules (Saxe et al., 2022). The nonlinearity is achieved by gating on or off portions of the network computation for different inputs. Formally, let $\Gamma$ denote a directed graph with nodes $\mathbf{V}$ and edges $\mathbf{E}$. Each node $v \in \mathbf{V}$ represents a layer of neurons with activity $h_v \in \mathbb{R}^{|v|}$ where $|v| \in \mathbb{N}$ denotes the number of neurons in the layer. Each edge $e \in \mathbf{E}$ connects two nodes with a weight matrix $W_e$ of size $|t(e)| \times |s(e)|$ where $s, t : \mathbf{E} \to \mathbf{V}$ return the source and target nodes of the edge, respectively. It is also helpful to generalize this notion of edges to paths $p$, which are a sequence

of edges $W_p$. Thus, paths also connect their source node $s(p)$ to their target node $t(p)$. We denote the portion of a path $p$ preceding a particular edge $e$ as $\bar{s}(p,e)$ and the portion of the path subsequent to the edge as $\bar{t}(p,e)$ such that $W_p = W_{\bar{t}(p,e)} W_e W_{\bar{s}(p,e)}$. We collect nodes with only outgoing edges into the set $\text{In}(\Gamma) \subset \mathbf{V}$ and call them input nodes. Similarly, output nodes only have incoming edges and are collected in the set $\text{Out}(\Gamma) \subset \mathbf{V}$. Activity is propagated through the network for a given datapoint which specifies values $x_v \in \mathbb{R}^{|v|}$ for all input nodes $v \in \text{In}(\Gamma)$. When concatenated the entire data point is denoted as $x$. The activity of each subsequent layer is then given by:

$$h_v = g_v \sum_{q \in \mathbf{E}:t(q)=v} g_q W_q h_{s(q)} \tag{1}$$

Here $g_v$ is a node gate and $g_q$ is an edge gate. Thus, the gating variables modulate the propagation of activity through the network by switching off entire nodes ($g_v$) and edges between nodes ($g_q$).

We train the GDLN to minimize the $L_2$ loss averaged over the full dataset of $N$ datapoints using gradient descent. The $i$-th datapoint then is a triple specified by the input $x^i$, output labels $y^i$ and specified gating structure $g^i$, where $y_v^i \in \mathbb{R}^{|v|}$ for $v \in \text{Out}(\Gamma)$.

$$L(\{W\}) = \frac{1}{2N} \sum_{i=1}^{N} \sum_{v \in \text{Out}(\Gamma)} ||y_v^i - h_v^i||_2^2 \tag{2}$$

Note that in Saxe et al. (2022) the gates are treated as part of the dataset. As the main results of this work do not depend on how we find the gates for our GDLN, we follow this precedent. However, the ease of finding the gates is an important consideration for the usability of the paradigm we introduce here. Thus, in Appendix B we discuss a very simple clustering approach to finding the gates from a ReLU network.

If we denote $\mathcal{P}(e)$ as the set of paths which pass through edge $e$, $\mathcal{T}(v)$ as the set of paths terminating at node $v$ and $\tau = \frac{1}{N\epsilon}$ then the update step for a single weight matrix in the GDLN is:

$$\tau \frac{d}{dt} W_e = -\frac{\partial L(\{W\})}{\partial W_e} = \sum_{p \in \mathcal{P}(e)} W_{\bar{t}(p,e)}^T \left[ \Sigma^{yx}(p) - \sum_{j \in \mathcal{T}(t(p))} W_j \Sigma^x(j,p) \right] W_{\bar{s}(p,e)}^T \ \forall \ e \in \mathbf{E} \tag{3}$$

Thus, the update to a weight matrix $W_e$ is determined by the error at the end of a path which it contributes to $[\Sigma^{yx}(p) - \sum_{j \in \mathcal{T}(t(p))} W_j \Sigma^x(j,p)]$ summed over all paths it is a part of $p \in \mathcal{P}(e)$. Notably, all dataset statistics which direct learning are collected into the correlation matrices:

$$\Sigma^{yx}(p) = \frac{1}{N} \sum_{i=1}^{N} g_p^i y_{t(p)}^i x_{s(p)}^{iT}; \qquad \Sigma^x(j,p) = \frac{1}{N} \sum_{i=1}^{N} g_j^i x_{s(j)}^i x_{s(p)}^{iT} g_p^i \tag{4}$$

These dataset statistics depend on the gating variables specific to a datapoint $g^i$, indicating that each path sees its own effective dataset determined by the network architecture. From this perspective, each path resembles the gradient flow of a deep linear network (Saxe et al., 2014; 2019) which have been shown to exhibit nonlinear learning dynamics observed in general deep neural networks (Baldi & Hornik, 1989; Fukumizu, 1998; Arora et al., 2018; Lampinen & Ganguli, 2019). The final step then is to use the change of variables employed by the linear network dynamics (Saxe et al., 2014; 2019). Assuming the effective dataset correlation matrices are mutually diagonalizable (such that $V$ is the same matrix for $\Sigma^{yx}$ and $\Sigma^x$), we write the correlation matrices and pathway weights in terms of their singular value decompositions:

$$\Sigma^{yx}(p) = U_{t(p)} S(p) V_{s(p)}^T; \quad \Sigma^x(j,p) = V_{s(j)} D(j,p) V_{s(p)}^T; \quad W_e(t) = R_{t(e)} B_e(t) R_{s(e)}^T \tag{5}$$

where $S(p)$ and $D(p)$ are diagonal singular value matrices, $B_e(t)$ are the new dynamic variables, $R_v$ satisfies $R_v^T R_v = I$ for all $v \in \mathbf{V}$, and for input and output nodes $R_{s(p)} = V_{s(p)}$ and $R_{t(p)} = U_{t(p)}$, respectively. For $R_{s(p)} = V_{s(p)}$ and $R_{t(p)} = U_{t(p)}$ we assume that the pathway mappings align to the same singular vectors which diagonalise their effective datasets and apply the same change of variables to the pathway weights. This assumption has been used successfully for feature learning linear networks previously (Saxe et al., 2014; 2019; Jarvis et al., 2023) and the trend for networks to align in this manner has be termed the "silent alignment effect" (Atanasov et al., 2021). This change of variables removes competitive interactions between singular value modes along a path such that

the dynamics of the overall network is described by summing several "1D networks" (one for each singular value) resulting in the "neural race reduction" (Saxe et al., 2022):

$$\tau \frac{d}{dt} B_e = \sum_{p \in \mathcal{P}(e)} B_{p \setminus e} \left[ S(p) - \sum_{j \in \mathcal{T}(t(p))} B_j D(j, p) \right] \ \forall \, e \in \mathbf{E} \qquad (6)$$

From this reduction we note that the update to a layer of weights is proportional to the input-output singular values of the effective dataset along a pathway it contributes to, and the number of these pathways (shown by the summation over $p \in \mathcal{P}(e)$). Thus, we can summarize the assumptions of the GDLN paradigm from Saxe et al. (2022) and note that we do not introduce any new assumptions:

**Assumption 2.1.** The assumptions of the GDLN paradigm are:

1. The dataset correlation matrices are mutually diagonalizable.

2. The neural network weights align to the singular vectors of the dataset correlation matrices.

Importantly, Assumption 2.1.1 is necessary for the full tractability and interpretability offered by the GDLN framework. However, if it is violated we can still continue the derivation to a dynamics reduction which is the same as the neural race reduction (Saxe et al., 2022) but includes an additional matrix multiplication, which is insightful and likely to be enough for many cases. Assumption 2.1.2 is stricter, however in the present context this is the same as assuming the network is successfully feature learning. We elaborate more on both assumptions in Section A and note that Section 6 depicts the utility of our framework in a case where the alignment of the singular vectors is not guaranteed.

## 3 THE RECTIFIED LINEAR NETWORK (RELN)

We denote the output of a GDLN for the $i$-th data point $x^i$ in a dataset of $N$ data points as $GDLN(W(t), G(x^i, t), x^i)$ where $W(t)$ is the set of weights and $G(x^i, t)$ is the set of gates that are dependent on the data point. Both the gates and weights can be time dependent, however in this section and Sections 4 and 5 the gating structure is constant from the beginning. Thus, $G(x^i, t) = G(x^i, 0) = G(x^i) \, \forall \, t \in \mathbb{R}^+$ in these sections. We denote the ReLU network's output as $ReLU(\overline{W}(t), x^i)$ with $\overline{W}(t)$ as the set of weights. We formally define a ReLN as:

**Definition 3.1. Rectified Linear Network (ReLN)**: The GDLN with $G(x^i) = G^*(x^i)$ such that $GDLN(W(t), G^*(x^i), x^i) = ReLU(\overline{W}(t), x^i) \, \forall \, t \in \mathbb{R}^+$ and $\forall \, i \in \{0, 1, ..., N\}$.

This definition is not specific to the tasks considered in this work and it is *always* possible to obtain a ReLN for a given ReLU network and task. Letting $\overline{h}_j = \sum_{k=0}^{K} \overline{W}_{jk} x_k^i$ be the pre-activation of the $j$-th hidden neuron in our ReLU network for any given data point $x^i$ ($K$ is the number of input features), then the post-activation from the ReLU activation can be written as: $\sigma(\overline{h}_j) = step(\overline{h}_j)\overline{h}_j$. Thus, in a GDLN with exactly the same architecture we could gate the corresponding neuron with $G(x^i)_j = step(\overline{h}_j)$. In other words we could gate the GDLN hidden neurons based on the activity of the ReLU network explicitly. This extends to multiple hidden layers as well. However, this is not desirable and demonstrates the extremity of the ReLN framework. Instead, we aim to find as few unique gating patterns $G(x^i)$ across the dataset as possible – exposing the functional "modules" of the network. We now illustrate this approach with an example.

**Transition to nonlinear separability.** ReLU networks have the ability to perform nonlinear tasks. Yet when faced with a linearly separable problem, they can also possibly learn a linear decision boundary. Here we examine a task which permits both of these possibilities, to examine the implicit bias in gradient descent. As depicted in Figure 2a, the task has XoR structure in its first two dimensions; and is linearly separable with margin $2\Delta$ in the third. Hence for $\Delta = 0$, the task is not linearly separable, but for $\Delta > 0$ it is. To identify a ReLN on this task, we consider two intuitive gating structures. The first attempts to exploit the linearly separable structure and contains two pathways, one active for positive examples and one active for negative examples, as depicted in Figure 2b. The second GDLN (Figure 2c) contains four pathways, each active on exactly one example. This GDLN can solve the task even when $\Delta = 0$, because it uses a different linear model for each datapoint.

Given the problem, these gating structures are intuitive. It remains to show that they behave like a ReLU network trained on the same task. Figure 2e shows the loss dynamics of each GDLN and for

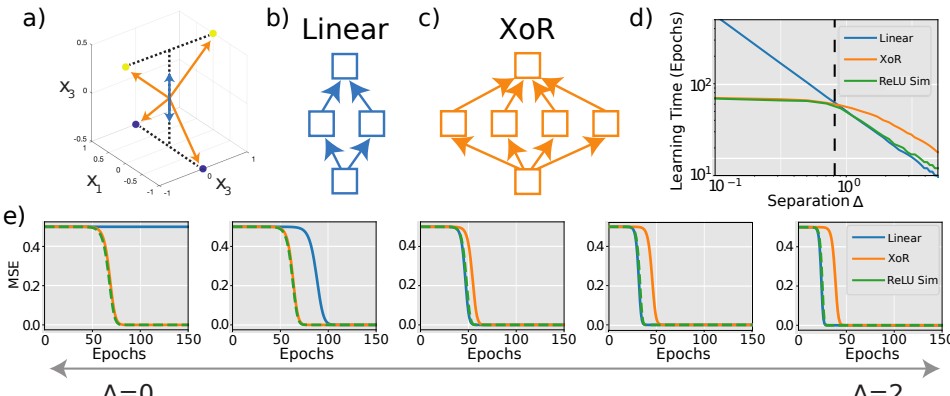

Figure 2: Dynamics of ReLU networks during transition to nonlinear separability. (a) A simple dataset which has XoR structure in the first two dimensions, and is linearly separable with margin $2\Delta$ in the third. (b) GDLN exploiting linear structure. The network contains two pathways, one gated on only for positive examples and one gated on only for negative examples. Blue arrows in panel (a) depict ReLU weight directions that achieve this gating. (c) GDLN exploiting XoR structure. The network contains four pathways, each active on exactly one example. Orange arrows in panel (a) depict ReLU weight directions that achieve this gating. (d) Time to learn to a fixed criterion (loss=.2) calculated analytically for GDLNs with linear and XoR gating structure (blue and orange, respectively), and in simulation of ReLU networks (green). The ReLU network behaves like the faster of the two gated networks. Which gating structure is fastest changes at $\Delta = \sqrt{2/3}$ (grey dashed). (e) Analytical loss trajectories for the gated networks, and simulated ReLU networks for several values of $\Delta$. The full trajectories of the simulation match the faster of the gated networks. *Parameters: learning rate $1/\tau = .4$, $N_h = 128$ hidden units, initialization variance $4 \cdot 10^{-8}/N_h$.*

a 128 hidden unit ReLU network trained on the same task, for a range of values of $\Delta$. We see that across the spectrum of $\Delta$s, one of these GDLNs always closely matches the ReLU network dynamics – which is the ReLN in that setting. Figure 2d shows the learning time of these networks as a function of $\Delta$, showing that ReLU networks behave like the faster GDLN (the ReLN).

Because of the simple structure in these ReLNs, it is possible to solve their dynamics analytically (see Appendix C). These dynamics yield a prediction for the behaviour of the ReLU network. Using these solutions, we can calculate the value of $\Delta$ when the speed of the linear GDLN crosses over that of the XoR ReLN. This crossover point is $\Delta = \sqrt{2/3}$ (Figure 2d grey dashed line), which marks the approximate point at which a ReLU network switches from a linear decision boundary to a nonlinear boundary (see Appendix C). Perhaps surprisingly, this analysis shows that the ReLU network transitions to a nonlinear boundary well before the problem becomes nonlinearly separable.

Hence this example demonstrates how ReLNs with appropriate gating can describe the behaviour of ReLU networks. When the ReLN has a simple structure, its greater tractability permits analytical solutions that closely describe ReLU dynamics on the same task. Finally, this example shows that while gradient-trained ReLU networks do have a bias toward linear decision boundaries when the margin is large, they can adopt nonlinear decision boundaries even when a problem is linearly separable.

## 4 EMERGENT STRUCTURED MIXED SELECTIVITY IN RELU NETWORKS

Having introduced ReLNs in Section 3 we now apply this theory to understand the inductive biases of ReLU networks in a more challenging setting, summarized in Figure 3. Similar to prior work we create a task where the network is provided a object index as input and required to produce a set of corresponding properties for the object (Saxe et al., 2019; Braun et al., 2022), however we include a context feature to the input similar to previous connectionist models of controlled semantic cognition (Rogers et al., 2004) and more recently in multi-task learning (Sodhani et al., 2021). Each item is queried with a one-hot representation and one-hot context feature, such that all items are present in each context, forming the input matrix $X$ ($X_c$ denotes the contextual portion of the dataset). The outputs, corresponding to predicted features of the items, then impart structure into the dataset based on the similarity of items. For example, the first set of output labels (rows) form a hierarchy structure.

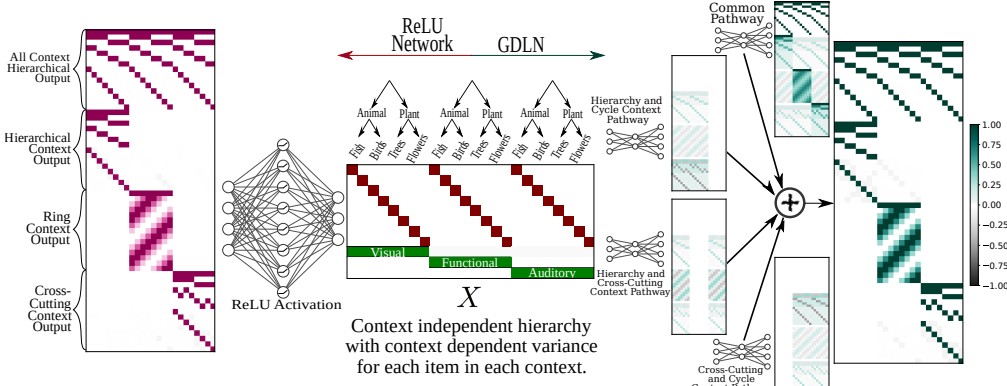

Figure 3: Dataset used to train the ReLU network (left) and ReLN architecture used to imitate it (right). Inputs (middle matrix) are created by appending a one hot vector encoding object identity to a one hot vector encoding context such that each item appears in all contexts. Target outputs (left and right matrices) contain some context-independent (top block) and some context specific properties (bottom three blocks). These datasets broadly follow a hierarchical structure across items (hierarchical tree depicted in the middle over input datapoint along the columns), but with some variation in each context-specific block. All structures are taken from Saxe et al. (2019). The analysis in this work shows that the ReLU network dynamics arise from four implicit modules, made explicit by the ReLN pathways towards the right, which receive different subsets of inputs and generate different subsets of outputs. Together these graded mixed-selective pathways couple together to produce the correct output labels for each object. While each context-specific pathway is only on in two contexts (blocks of columns) they still produce labels for all three context-specific parts of the output space (blocks of rows). This creates errors which other pathways learn to remove. If this fine balance of activity between pathways is broken then errors will be incurred.

This block of features should be activated regardless of the queried context. In contrast the three other blocks of output labels need to be activated only in one of the contexts. This requires contextual control and a nonlinear network mapping to learn. We train the networks with full-batch gradient descent and $L2$ loss (Equation 2) to perform this task. We use a linear output layer, assume the model is over-parametrized (a pathway needs to have at least $h$ hidden nodes to learn a rank $h$ effective dataset) and do not regularize or bias the networks towards context specificity in its hidden neurons. The only difference between the two networks is that the gating of the ReLN's hidden neurons is explicit, while the ReLU network learns the gating pattern using the nonlinearity. Remarkably, in this setting the gating of the ReLN is constant from the start.

To obtain the closed-form dynamics of the ReLN, and by extension ReLU network, it is necessary to determine the effective datasets of each pathway through the network. We identify four pathways which imitate functional modules in the ReLU network. The first "common" pathway is composed of hidden neurons which are always active, forming a linear subnetwork responsible for learning the average activation across the dataset. While this pathway does not use context for gating, it still uses this as a feature similar to a bias variable. The three remaining pathways are all active for two out of the three contexts and collectively learn the residual of the common pathway. All pathways map onto the full output space. The ReLN and corresponding pathways are depicted on the right of Figure 3. Figure 4(a) shows the loss dynamics of the ReLU network compared to the ReLN and the predicted dynamics with samples of the ReLU and ReLN outputs at certain points during training[1]. We see excellent agreement across all dynamics and outputs. Further, comparing Figure 4(c) which show the Multi-dimensional Scaling (MDS) plots of the ReLU network and ReLN respectively, we see near exact correspondence between the relative positions of the latent representations of the dataset across both architectures. The derivation of the dynamics using the race reduction from Section 2 is presented in full in Appendix F.

By interpreting the corresponding ReLN, we find that there are **no neurons** in the ReLU network which are context specific. Instead, the ReLU network employs a strategy of mixed selectivity where neurons are active across multiple contexts. These pathways are still doing feature learning, however, as evidenced by the effective datasets which we uncover and the corresponding singular

---

[1]Code using the Jax Library (Bradbury et al., 2018) is at `https://tinyurl.com/gdln-reln`.

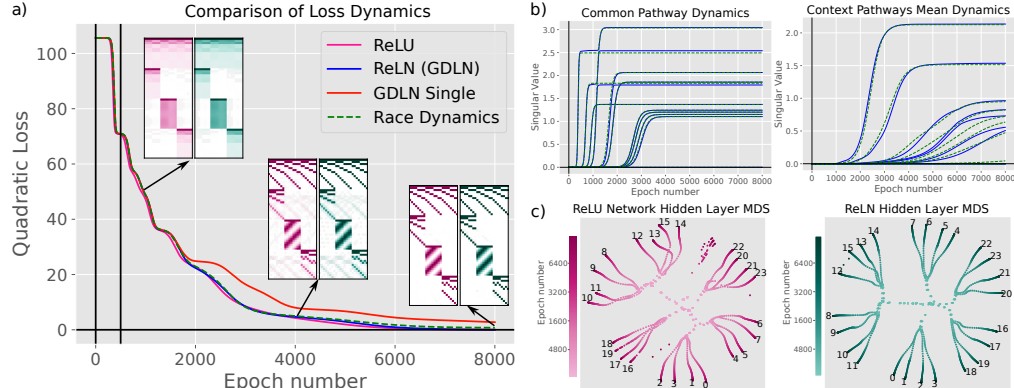

Figure 4: Summary of results for the ReLN imitating a ReLU network on a contextual nonlinear task. (a) Comparison of loss trajectories between the ReLU, empirical and predicted ReLN, and alternate GDLN which does not imitate the ReLU network loss trajectory. We find that a GDLN (here called "GDLN Single") which has contextual pathways active for individual contexts is unable to imitate the loss trajectory of the ReLU network. This is used in Proposition 4.2 to prove the uniqueness of the ReLN which we have identified. Example outputs from the ReLU network and ReLN are also shown and we see exact agreement between these output samples. (b) Singular value dynamics for the ReLN architecture using the neural race reduction dynamics. (c) Multi-dimensional Scaling which compares the relative latent representations of the ReLU network and ReLN over time. Both architectures demonstrate an equivalent latent representation at all points in time.

value trajectories for each summarized in Figure 4(b). Finally, Figure 4 only shows that the loss of the ReLN matches that of the ReLU network, which is a weaker condition than producing the same output at all times, as in Definition 3.1. A natural question may also emerge on whether another ReLN exists which provides the same dynamics. To answer this, we present Proposition 4.2 which proves that the loss trajectory of this ReLN is unique (accounting for symmetric gating patterns such as permuting hidden neurons (Simsek et al., 2021; Martinelli et al., 2023)). In addition, the ReLN which emerges is the fastest learner of all GDLN architectures implementable by a ReLU network, winning the "neural race" (Saxe et al., 2022). Thus, while the ReLU network is doing "slow" feature learning, it does so by utilizing the most efficient network substructures. The structure emerges from feature learning, while mixed-selectivity and node reuse remains as a strategy for learning speed, commonly found in the "fast" lazy learning regime (Jacot et al., 2018; Geiger et al., 2020). Importantly, mixed-selectivity is an established concept within the neuroscience literature (Anderson, 2010; Rigotti et al., 2013), however this balancing of learning speed and feature learning reflects a potential mechanism for its emergence. Before we present Proposition 4.2 we first require Lemma 4.1 which limits the possible gating strategies implementable by a ReLU network with a single hidden layer on this dataset and drastically compresses the possible set of GDLN architectures which could be potential ReLNs. The full proof can be found in Appendix D.

**Informal Lemma 4.1** Any gating strategy employed by a trained ReLU network with a single hidden layer on the input $X$ (as defined in Section 4) can be implemented by a GDLN with linear modules that gate based on the context features in isolation or are only active for a single datapoint.

**Lemma 4.1.** *Given the definition of GDLNs and ReLU Networks, for any gating pattern implementable by the ReLU network there is $G(x_c)$ such that $G(x_c)_j = step(\overline{h}_j)$ or $G(x)_j = \mathbb{1}(x) \, \forall \, j \in \{0, 1, ..., H\}$ where $\overline{h}$ is the pre-activations of the ReLU network's hidden layer with $H$ neurons and $x_c$ is the portion of the data point $x$ with contextual features.*

> **Proof Sketch:** We note that a ReLU network must use the same set of weights to perform the gating and forward propagation of information. Without loss of generality we then consider the simple case of two objects in two contexts forming a dataset of four datapoints. We show by contradiction that a hidden neuron is unable to gate inconsistently for both object and context, for example by being active for item 1 in context 1 and item 2 in context 2 while being off on the remaining datapoints. This means all hidden neurons must gate consistently using either context or object features. We then consider a strategy where a neuron gates based on an item's features. In this case the readout from the neuron will produce outputs across the entire label

space agnostic to the requested context as it will be activated by all context features. Thus, it is unable to perform the nonlinear mapping from context to output. Consequently the only viable gating strategy for the network is to partition the dataset using the context features in isolation or allocate subnetworks to each datapoint.

With the search space of the ReLN within the space of GDLNs reduced we may prove that the loss trajectory of the ReLN in this section is unique. The full proof of Proposition 4.2 can be found in Appendix E. Note that $L_{GDLN}$ and $L_{ReLU}$ are the loss of the GDLN and ReLU networks respectively.

**Informal Proposition 4.2** The ReLU network in this section: 1) has a unique corresponding ReLN which imitates the loss dynamics and output at all times, 2) finds the neural race winning strategy of mixed selectivity.

**Proposition 4.2.** *There is a unique $G^*(X)$ (up to symmetries such as permutations) such that $L_{GDLN}(W(t), G^*(X)) = L_{ReLU}(\bar{W}(t)) \ \forall \ t \in \mathbb{R}^+$ and for alternative gating strategies $G'(X)$ then $L_{GDLN}(W(t), G^*(X)) < L_{GDLN}(W(t), G'(X))$*

**Proof Sketch:** We first make use of a generalization of the Cauchy Interlacing Theorem to non-square matrices, closely following the strategy from Thompson (1972). We show that if an output label or input feature is removed from the dataset, the first singular value of the input-output correlation matrix ($\Sigma^{yx}$) can only decrease. Since the singular values of the correlation matrix alone determine the time-course of learning in linear networks and pathways (Saxe et al., 2014; 2022) this means a network can only learn slower when removing these input or output dimensions. Thus, pathways using all available features will win the neural race, consistent with our ReLN architecture, and any other strategy would result in a clear deviation from the ReLU loss trajectory. We then consider whether gating along datapoints and not features could provide an alternative architecture. Here we note that the output of the common pathway is element-wise positive. Thus, removing a datapoint with only positive terms can only decrease the correlation calculation. Thus, the common pathway which uses all features and is active for all datapoints is the fastest possible module the network could implement. Lastly, we consider if alternative gatings along the context specific pathways exist. Using Lemma 4.1 we note that the network can only implement gates which partition the input consistently based on the contexts or activate separate modules for each datapoint. For the latter case, once the common pathway has finished learning there is no remaining correlation between the context features and residual. Thus when a single datapoint is allocated to a single module all activity would need to be learned by the item feature. In our dataset, removing all other items in a context would then be the same as removing their individual one-hot input features. From the Cauchy Interlacing Theorem, we know this can only slow learning. Thus, there are two remaining strategies which gate consistently based on context, making a proof by exhaustion feasible, and so we simply simulate both and plot the trajectories in Figure 4. Clearly the architecture with pathways active in two contexts learns faster and matches the loss trajectory of the ReLU network. Thus, only one ReLN in the space of implementable GDLNs matches the ReLU network loss trajectory, and this is the fastest possible ReLN which wins the neural race.

## 5   THE EFFECT OF ADDITIONAL CONTEXTS

Having identified the inductive bias towards mixed selectivity in ReLU networks we now consider whether this trend continues as we increase the number of contexts. Once again we construct a ReLN which imitates the ReLU networks and identify the effective datasets for each of its pathways. In this case we consider datasets with three, four and five contexts. Similar to Section 3, the output labels contain a block with hierarchical structure which is active for all contexts. This is appended with context specific blocks which are active for a single context. Here we use a hierarchy structure for all contexts to isolate the effect of scaling the number of contexts. Additionally, with this repeated structure it is possible to continue the derivation of the dynamics beyond the neural race reduction and obtain fully closed form equations for the singular value trajectories of the linear pathways. The derivations for the three, four and five context cases are presented in Appendix F and G. We plot the closed form singular value trajectories and corresponding loss curves for each context in Figure 5 and once again see excellent agreement. Moreover we find that the trend identified in Section 3 continues, as the ReLN which imitates the ReLU network in each case uses a linear pathway that is active for all

contexts. The ReLN then has $C$ context specific pathways which are active for $C - 1$ contexts. Thus, as the number of contexts increases, so too does the bias towards node reuse and mixed selectivity.

Figure 5: Effect of increasing the number of contexts on mixed selectivity: Summary of the loss trajectory (first row) for the ReLU network and corresponding ReLN as the number of contexts increases (columns). Due to the symmetry of the task we are also able to continue the derivation of the ReLN dynamics beyond the neural race reduction and obtain closed form trajectories for the networks' singular values. The singular value trajectories for the common pathway (second row) and mean trajectories for the contextual pathways (bottom row) match simulations exactly. Consequently we can in closed form derive the loss trajectory for each architecture and see perfect agreement to both the ReLN and ReLU networks. All together these results demonstrate that as the number of contexts increase, so too do the number of contexts a pathway is active for.

## 6 THE EFFECT OF ADDITIONAL HIDDEN LAYERS

Finally, we consider the effect of adding a second hidden layer with ReLU activation to the network. Once again we are able to identify the corresponding GDLN which imitates the network. This GDLN architecture is summarized in Figure 6. We see that in this case the ReLU network does not use the first layer for gating and instead all neurons are active for all datapoints. However, this layer does still benefit the network as it enables it to gate using a combination of item and context variables in the *second* hidden layer. This removes the constraint from Lemma 4.1 and allows a wider range of gating strategies. However, we see that the gating strategy of the network remains relatively similar as the contextual pathways are active in two out of the three contexts. Similarly the common pathway remains. Thus, the trend towards mixed-selectivity and node reuse remains with the addition of another hidden layer, which is then used as a linear layer.

In Figure 6b we plot the trajectories from 100 runs of the ReLU network and GDLN respectively. We note that there is now variance in the trajectories of both networks, which was not the case for the single-hidden-layer architectures. This is due to an inconsistency in the timing at which the network singular vectors align with the singular vectors of the dataset correlation matrices. As a consequence there is variance in the loss trajectories as the contextual pathways do not align perfectly to a consistent set of singular vectors from early in training (violating Assumption 2). We see that the GDLN is still informative about the behaviour of the ReLU network. In Figure 6 we find the loss trajectories closest to the mean trajectory for the ReLU network and GDLN separately. We then plot these stereotypical loss trajectories for comparison and plot the faded remaining trajectories around this. We see close agreement between the ReLU network and GDLN stereotypical trajectories, but

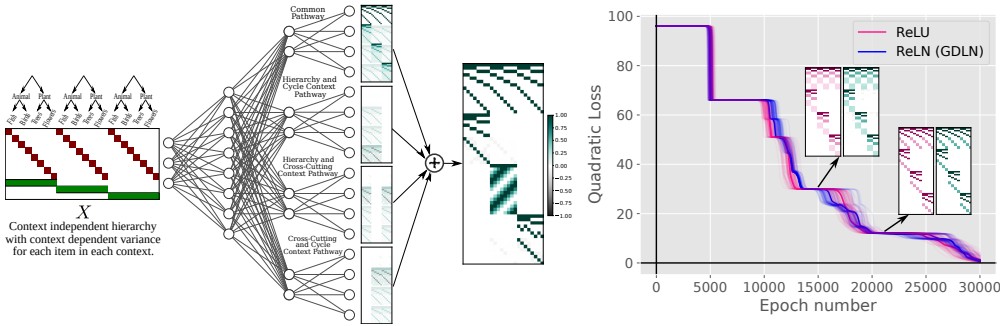

Figure 6: Summary of architecture and results for the addition of depth: With the addition of depth the ReLU network is no longer limited to gating consistently using context variables. However, we find that the network still uses a common pathway and contextual pathways active in two contexts. The first layer now is completely shared across all contexts and does not use the nonlinearity. This architecture is imitated by the ReLN depicted on the left. We note that there is now variance between runs of the ReLU network and the ReLN. Interestingly the ReLN shares the same characteristics in variance as the ReLU network. We plot 100 runs of each architecture and highlight one trajectory for the ReLU network and ReLN which could be considered stereotypical (using the sum of L2 distances across all timepoints). Overall the ReLN still provides clear insight into the ReLU network and its inductive bias towards mixed selectivity and node reuse.

we also see the same variance profiles across the various runs. We note that pathways within the GDLN become active in stages. For example, the common pathway align around epoch 500, while the contextual pathways only appear at epoch 12000. The relative timing of the drops in loss and positions of the plateaus in loss are all consistent following the alignment. Moreover, when comparing samples from both networks at plateaus in the loss trajectories we see exact agreement between the outputs. In summary, while the addition of a hidden layer allows the ReLU to have a more intricate gating structure, this comes with an additional difficulty of perfectly aligning to the dataset singular vectors. Thus, the ReLU networks and GDLNs fail to learn the exact optimal features on all runs.

## 7 DISCUSSION

In this work we have introduced the Rectified Linear Network which is a GDLN designed to imitate the gating structure of a specific ReLU network. We have used this to provide a theory of feature learning in finite ReLU networks and identified an implicit bias towards structured mixed selectivity which results due to node reuse. Our approach has several limitations. Most notably, while some GDLNs can imitate any ReLU, a reduction based on this idea will only be insightful if the resulting GDLN has few pathways. Otherwise, the many pathways of the GDLN become as hard to interpret as the many neurons in a ReLU network. A second limiting factor is how easily identifiable the gating structure is. As a first step towards addressing this we propose a very simple clustering based algorithm in Appendix B which we show is able to retrieve the correct modules from a ReLU network in the setup of Section 4. In this appendix we also briefly touch on the connection to meta-learning and polysemanticity (neurons being activated by multiple semantic unit or features) (Olah et al., 2017a; Lecomte et al., 2024), and promote these as directions of future work.

In the reasonably complex yet structured cases we examine here, we have shown that subnetworks emerge within a ReLU network due to the learning speed benefits offered by mixed-selectivity. Moreover, we find that the bias towards node reuse increases with the addition of more contexts and greater depth. Thus, our findings are in agreement with prior empirical and theoretical findings on disentanglement which demonstrate that neural networks are not implicitly biased towards dis-entangled representations that select for semantic factors of variation in the input (Locatello et al., 2019; Michlo et al., 2023). However, we find that mixed selectivity of neural representations does not imply that the network is not learning structured features. Instead the network exploits reusable structure between contexts, forming functional modules which we make explicit with pathways in the ReLNs. This demonstrates that node reuse and modularity are not as incongruent as previously thought in both machine learning (Andreas et al., 2016; Andreas, 2018; Masoudnia & Ebrahimpour, 2014) and neuroscience (Anderson, 2010; Rigotti et al., 2013) and we shed light on the competing factors influencing their emergence from learning.

ACKNOWLEDGMENTS

We thank Joseph Warren and Mingda Xu for useful discussions and feedback on this work. For the purpose of Open Access, the author has applied a CC BY public copyright license to any Author Accepted Manuscript version arising from this submission. A.S. is supported by the Gatsby Charitable Foundation and Sainsbury Wellcome Centre Core Grant (219627/Z/19/Z). D.J. is a Google PhD Fellow mentored by Gamaleldin F. Elsayed and a Commonwealth Scholar. A.S. and B.R. are CIFAR Azrieli Global Scholars in the Learning in Machines & Brains program. Computations were performed using a Nvidia GTX 1080 from the High Performance Computing infrastructure provided by the Mathematical Sciences Support unit at the University of the Witwatersrand.

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

## A    LINEAR NEURAL NETWORKS BACKGROUND

In this section we review the deep linear neural network theory paradigm. Importantly, we do not review other theories such as the Neural Tangent Kernel (Jacot et al., 2018) which linearize neural networks and establish the regimes when this linearization is valid. These theories are highly insightful in their own right and have shed light on the training of various neural networks in general (Jacot et al., 2018; Bietti & Mairal, 2019; Wang et al., 2022). However, we do not build directly on these works and so limit the scope of this review to the linear neural networks paradigm (Saxe et al., 2014; 2019) in Appendix A.1 and GDLNs (Saxe et al., 2022) in Appendix A.2.

### A.1    DEEP LINEAR NEURAL NETWORKS

The primary theoretical strategy in this work is to calculate the training dynamics of linear neural networks or modules formed from these networks. A first important distinction is between deep and shallow linear networks. While deep linear networks can only represent linear input-output mappings, the dynamics of learning change dramatically with the introduction of one or more hidden layers (Fukumizu, 1998; Saxe et al., 2014; 2019; Arora et al., 2018; Lampinen & Ganguli, 2019), and the learning problem becomes non-convex (Baldi & Hornik, 1989). They therefore serve as a tractable model of the influence of depth specifically on learning dynamics, which prior work has shown to impart a low-rank inductive bias on the linear mapping (Huh et al., 2021). The exact solutions to the dynamics of learning from small random weights in deep linear networks have been derived in Saxe et al. (2014; 2019) and as a result it is possible to obtain the full learning trajectory analytically for a number of representative tasks.

To review this paradigm consider a linear network with one hidden layer computing the predicted output $\hat{Y} = W^2 W^1 X$ in response to an input batch of data $X$, with $P$ datapoints, and trained to minimize the quadratic loss using gradient descent:

$$L(W^1, W^2) = \sum_{i=1}^{P} \frac{1}{2} ||Y_i - W^2 W^1 X_i||_2^2$$

This gives the learning rules for each layer with learning rate $\epsilon$ as:

$$\Delta W^1 = \epsilon P W^{2^T}(\Sigma^{yx} - W^2 W^1 \Sigma^x); \qquad \Delta W^2 = \epsilon P(\Sigma^{yx} - W^2 W^1 \Sigma^x) W^{1^T}$$

where $\Sigma^x$ is the input correlation matrix and $\Sigma^{yx}$ is the input-output correlation matrix defined as:

$$\Sigma^x = \frac{1}{P} X X^T; \qquad \Sigma^{yx} = \frac{1}{P} Y X^T$$

These equations can be derived for a batch of data using the linearity of expectation as follows:

$$
\begin{aligned}
\Delta W^1 &= \epsilon \frac{d}{dW^1} L(W^1, W^2) \\
&= \epsilon \frac{d}{dW^1} \sum_{i=1}^{P} \frac{1}{2} (Y_i - W^2 W^1 X_i)^T (Y_i - W^2 W^1 X_i) \\
&= \epsilon \sum_{i=1}^{P} W^{2^T} (Y_i - W^2 W^1 X_i) X_i^T \\
&= \epsilon P \frac{1}{P} \sum_{i=1}^{P} W^{2^T} (Y_i - W^2 W^1 X_i) X_i^T \\
&= \epsilon P W^{2^T} \left( \frac{1}{P} \sum_{i=1}^{P} Y_i X_i^T - W^2 W^1 \frac{1}{P} \sum_{i=1}^{P} X_i X_i^T \right) \\
&= \epsilon P W^{2^T} (\Sigma^{yx} - W^2 W^1 \Sigma^x)
\end{aligned}
$$

$$\Delta W^2 = \epsilon \frac{d}{dW^2} L(W^1, W^2)$$

$$= \epsilon \frac{d}{dW^2} \sum_{i=1}^{P} \frac{1}{2} (Y_i - W^2 W^1 X_i)^T (Y_i - W^2 W^1 X_i)$$

$$= \epsilon \sum_{i=1}^{P} (Y_i - W^2 W^1 X_i)(W^1 X_i)^T$$

$$= \epsilon P \frac{1}{P} \sum_{i=1}^{P} (Y_i - W^2 W^1 X_i) X_i^T W^{1^T}$$

$$= \epsilon P \left( \frac{1}{P} \sum_{i=1}^{P} Y_i X_i^T - W^2 W^1 \frac{1}{P} \sum_{i=1}^{P} X_i X_i^T \right) W^{1^T}$$

$$= \epsilon P (\Sigma^{yx} - W^2 W^1 \Sigma^x) W^{1^T}$$

By using a small learning rate $\epsilon$ and taking the continuous time limit, the mean change in weights is given by:

$$\tau \frac{d}{dt} W^1 = W^{2^T} (\Sigma^{yx} - W^2 W^1 \Sigma^x); \qquad \tau \frac{d}{dt} W^2 = (\Sigma^{yx} - W^2 W^1 \Sigma^x) W^{1^T}$$

where $\tau = \frac{1}{P\epsilon}$ is the learning time constant. Here, $t$ measures units of learning epochs. It is helpful to note that since we are using a small learning rate the full batch gradient descent and stochastic gradient descent dynamics will be the same. Saxe et al. (2019) has shown that the learning dynamics depend on the singular value decomposition of the correlation matrices, where $u^\alpha$ and $v^\alpha$ are singular vectors grouped into the matrices $U$ and $V$ respectively. Further, $\delta_\alpha$ and $\lambda_\alpha$ are the singular values of $\Sigma^x$ and $\Sigma^{yx}$ respectively and grouped into the matrices $D$ and $S$. Then the correlation matrices decompose as:

$$\Sigma^{yx} = USV^T = \sum_{\alpha=1}^{|YX^T|} \lambda_\alpha u^\alpha v^{\alpha^T}; \qquad \Sigma^x = VDV^T = \sum_{\alpha=1}^{|XX^T|} \delta_\alpha v^\alpha v^{\alpha^T}$$

Here $|\cdot|$ denotes the rank of the matrix. To solve for the dynamics we require that $\Sigma^{yx}$ and $\Sigma^x$ are mutually diagonalizable such that the right singular vectors $V$ of $\Sigma^{yx}$ are also the singular vectors of $\Sigma^x$. We verify that this is true for the tasks considered in this work and assume it to be true for these derivations. We also assume that the network has at least $|YX^T|$ hidden neurons (the rank of $\Sigma^{yx}$ which determines the number of singular values in the input-output correlation matrix) so that it can learn the desired mapping perfectly. If this is not the case then the model will learn the top $n_h$ singular values of the input-output mapping where $n_h$ is the number of hidden neurons (Saxe et al., 2014). To ease notation for the remainder of this section we use $n_h$ to denote both the number of hidden neurons and rank of $\Sigma^{yx}$.

We now perform a change of variables using the SVD of the dataset statistics. The purpose of this step is to decouple the complex dynamics of the weights of the network, with interacting terms, into multiple one-dimensional systems. Specifically we set:

$$W^2 = U\overline{W}^2 R^T; \qquad W^1 = R\overline{W}^1 V^T$$

where $R$ is an arbitrary orthogonal matrix such that $R^T R = I$. Substituting this into the gradient descent update rules for the parameters above yields:

$$\tau \frac{d}{dt} W^1 = W^{2^T} (\Sigma^{yx} - W^2 W^1 \Sigma^x)$$

$$\tau \frac{d}{dt} (R\overline{W}^1 V^T) = R\overline{W}^2 U^T (USV^T - U\overline{W}^2 R^T R\overline{W}^1 V^T V D V^T)$$

$$\tau \frac{d}{dt} (R\overline{W}^1 V^T) = R\overline{W}^2 (SV^T - \overline{W}^2 \overline{W}^1 D V^T)$$

$$\tau \frac{d}{dt} \overline{W}^1 = \overline{W}^2 (S - \overline{W}^2 \overline{W}^1 D)$$

and

$$\tau\frac{d}{dt}W^2 =(\Sigma^{yx} - W^2W^1\Sigma^x)W^{1^T}$$

$$\tau\frac{d}{dt}(U\overline{W}^2R^T) =(USV^T - U\overline{W}^2R^TR\overline{W}^1V^TVDV^T)V\overline{W}^1R^T$$

$$\tau\frac{d}{dt}(U\overline{W}^2R^T) =(US - U\overline{W}^2\overline{W}^1D)\overline{W}^1R^T$$

$$\tau\frac{d}{dt}\overline{W}^2 =\overline{W}^1(S - \overline{W}^2\overline{W}^1D)$$

Here we have used the orthogonality of the singular vectors such that $V^TV = I$ and $U^TU = I$. Importantly, if the assumptions that the weight matrices align and the correlation matrices are mutually diagonalisable hold, then all matrices in the dynamics are now diagonal and represent the decoupling of the network into the modes transmitted from input to the hidden neurons and from hidden to output neurons. In practice we do not initialize the network weights to adhere to this diagonalisation and so it is not guaranteed that the matrices will be diagonal at initialization. However, empirically it has been found that the network singular values rapidly align to this required configuration (Saxe et al., 2014; 2019) and this has been termed the "silent alignment effect" (Atanasov et al., 2021). However, if this alignment does not occur fully then the weight matrices after the change of variables will not be diagonal. The alignment assumption corresponds to the assumption that the network is feature learning perfectly. While the need for perfect feature learning is strong, it is a valid start for the questions around feature learning considered in this work. If the assumption of the correlation matrices being mutually diagonalisable does not hold then the update equations simplify to:

$$\tau\frac{d}{dt}\overline{W}^1 = \overline{W}^2(S\hat{V}^TV - \overline{W}^2\overline{W}^1D) \qquad \text{and} \qquad \tau\frac{d}{dt}\overline{W}^2 = \overline{W}^1(S\hat{V}^TV - \overline{W}^2\overline{W}^1D)$$

where $\hat{V}$ now denotes the right singular vectors of $\Sigma^{yx}$ and are different to $V$ from $\Sigma^x$. We note that the remainder of the derivation requires both assumptions, however these two equations provide a valid dynamics reduction which could be sufficient for many interesting cases. If the weight alignment holds then this is especially true as $\hat{V}^TV$ would become a small interpretable matrix used to align the singular vectors from the correlation matrices. Thus, if we let $\lambda_\alpha$ be the $\alpha$-th mode of $S$ and define $\delta_\alpha$ similarly for D, $\omega_\alpha^1$ for $\overline{W}^1$ and $\omega_\alpha^2$ for $\overline{W}^2$ then we can write the individual mode dynamics as:

$$\tau\frac{d}{dt}\omega_\alpha^1 = \omega_\alpha^2(\lambda_\alpha - \omega_\alpha^2\omega_\alpha^1\delta_\alpha); \qquad\qquad \tau\frac{d}{dt}\omega_\alpha^2 = \omega_\alpha^1(\lambda_\alpha - \omega_\alpha^2\omega_\alpha^1\delta_\alpha)$$

In general $\omega^1$ and $\omega^2$ can be different but if they are initialized with small values then they will be roughly equal. We study this balanced setting and assume it to be true for all dynamics calculations in this work. Thus we let $\omega^1 = \omega^2$ and track the dynamics of an entire mode as $\omega_\alpha = \omega_\alpha^2\omega_\alpha^1$. Using the product rules this gives the separable differential equation (we drop the dependence on $\alpha$ now for notational convenience):

$$\tau\frac{d}{dt}\omega =\omega^1(\tau\frac{d}{dt}\omega^2) + (\tau\frac{d}{dt}\omega^1)\omega^2$$

$$\tau\frac{d}{dt}\omega =\omega^1(\omega^1(\lambda - \omega^2\omega^1\delta)) + (\omega^2(\lambda - \omega^2\omega^1\delta))\omega^2$$

$$\tau\frac{d}{dt}\omega =(\omega^1)^2(\lambda - \omega^2\omega^1\delta) + (\omega^2)^2(\lambda - \omega^2\omega^1\delta)$$

$$\tau\frac{d}{dt}\omega =\omega(\lambda - \omega\delta) + \omega(\lambda - \omega\delta)$$

$$\tau\frac{d}{dt}\omega =2\omega(\lambda - \omega\delta)$$

This is a separable differential equation and integrating to solve for $t$ then yields:

$$dt =\frac{\tau}{2}\frac{d\omega}{\omega(\lambda - \omega\delta)}$$

$$t =\frac{\tau}{2}\int_{\omega^0}^{\omega^f}\frac{d\omega}{\omega(\lambda - \omega\delta)}$$

$$t = \frac{\tau}{2\lambda} ln \frac{\omega^f(\lambda - \omega^0\delta)}{\omega^0(\lambda - \omega^f\delta)}$$

where $t$ is the time taken for the mode to reach a value $\omega(t) = \omega^f$ from the initial strength $\omega(0) = a^0$. By re-arranging the terms we can obtain the dynamics of the mode for all points in time:

$$\pi_\alpha(t) = \frac{\lambda_\alpha/\delta_\alpha}{1 - (1 - \frac{\lambda_\alpha}{\delta_\alpha \pi_0}) \exp(\frac{-2\lambda_\alpha}{\tau}t)} \tag{7}$$

All together this means that, given the SVDs of the two correlation matrices, the learning dynamics can be described explicitly as:

$$W^2(t)W^1(t) = UA(t)V^T = \sum_{\alpha=1}^{|YX^T|} \pi_\alpha(t)u^\alpha v^{\alpha T}$$

where $A(t)$ is the effective singular value matrix of the network's mapping and the trajectory of each singular value in $A(t)$ is described by $\pi_\alpha(t)$ which begins at the initial value $\pi_0$ when $t = 0$ and increases to $\pi_\alpha^* = \lambda_\alpha/\delta_\alpha$ as $t \to \infty$. From these dynamics it is helpful to note that the time-course of the trajectory is only dependent on the $\Sigma^{yx}$ singular values. Thus, $\Sigma^x$ affects the stable point of the network singular values but not the time-course of learning.

To make this setup more concrete, consider the example shown in Figure 7. For simplicity in this case we pick the input to be the identity matrix where each datapoint is a 1-hot vector. This 1-hot vector then depicts the index of an item in the dataset. The task then is for the network to produce a set of features or properties for each item. For example, when asked about the "Canary" item the network must predict that it can "Grow", "Move" and "Fly" but must not indicate that it has "Roots". Importantly, this dataset has a hierarchical structure defined by the output labels. For example, all items can grow, animals can move and plants have roots. Thus, the grow item depicts the top level of the hierarchy. Move and roots discriminate between animals and plants. Then each item has its own feature unique to it. This discriminates types of animals (canary vs salmon) and types of plants (oak vs rose).

The SVD of this dataset produces the singular vector matrices $U$ and $V^T$ depicting output and input "concepts" respectively. Due to the hierarchical nature of the dataset the input singular vectors discriminate between levels of the hierarchy. For example the second row of $V^T$ has positive elements to detect animals and negative elements to detect plants. The last two rows identify types of animals and types of plants. The output concepts then are the features corresponding to each level of the hierarchy. For example, the second column of $U$ which corresponds with the animal vs plant discriminator of $V^T$ has positive elements for the move attribute and negative elements for roots. Taken together $U$ and $V^T$ depict that animals can move and do not have roots. The $S$ matrix then depicts the strength of association from input to output concepts. It is important to note that these interpretable input and output singular vector matrices and the mode strengths arise purely from the dataset statistics.

Based on the dataset statistics we can then determine the linear neural network training dynamics in terms of the association it has between concepts - its mode strength when written in terms of the dataset SVD. Thus, we again assume that the singular vectors align very early on in training and it is sufficient to consider just the decoupled modes. If this is the case then the full neural network training dynamics can be obtained in closed-from using Equation 7. In this example we see near exact agreement between the closed-form predicted dynamics and the actual dynamics of training the neural network.

## A.2    GATED DEEP LINEAR NETWORKS

Saxe et al. (2022) extends the linear network dynamics framework above by incorporating a gating mechanism. One primary benefit of the gating is to add non-linearity to the network without needing to add activation functions. As a result the training dynamics for this model are also tractable. The

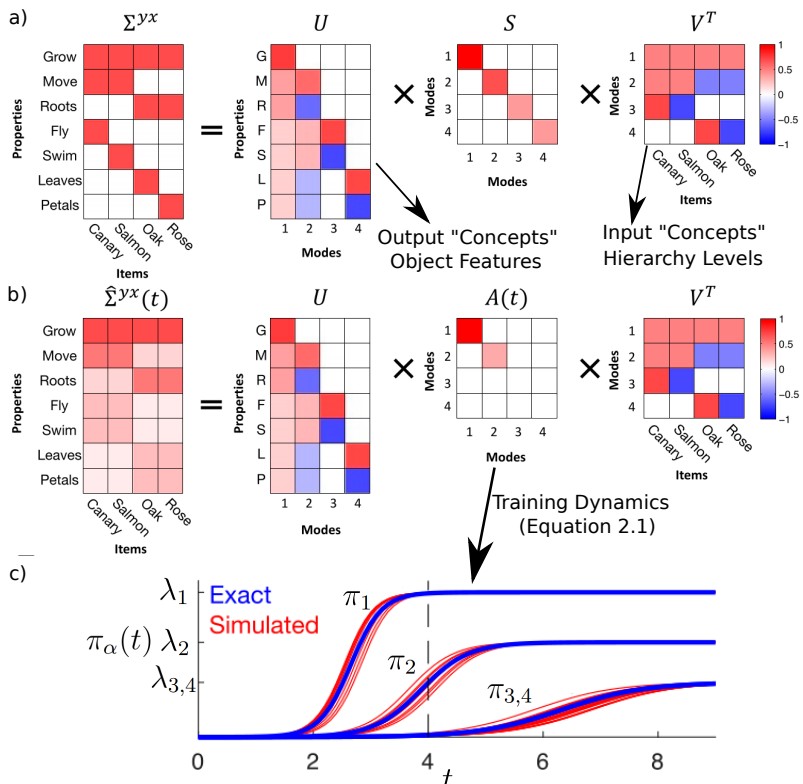

Figure 7: Example dataset and linear dynamics from Saxe et al. (2019). (a) Dataset with hierarchical structure defined by the output labels and the corresponding SVD. (b) The linear neural network decomposition in terms of the dataset SVD. $A(t)$ will develop over time to learn the association between modes from $S$. (c) The dynamics of the linear network mode strengths over time. Initially all modes begin at 0 but increase to their final values where they have learned the dataset statistics. Dataset inspired by Rogers et al. (2004), and the exact setting and dynamics are from (Saxe et al., 2019).

framework is known as the Gated Deep Linear Network (GDLN) and defines a dynamic neural architecture by using an "architecture graph" where a particular walk through the architecture graph provides a network instantiation which is only used for a subset of the training dataset. Every edge in the graph is a matrix of learnable network parameters and every vertex is a layer of neurons. Consequentially, vertices with only out-going edges correspond to the input layers of the network (a layer of neurons who's values are specified by the dataset, typically denoted as $x_i \in X$). Similarly, vertices with only in-coming edges correspond to output layers of the network (typically denoted as $\hat{y}_i$ which provides an approximation of some ground truth label $y_i$).

When presented with an input datum a set of gating variables $g_v$ and $g_e$ control the flow of information through the graph by switching off neurons within a layer ($g_v$) and edges in the graph ($g_e$). Thus, the pathway – sequence of operations – the datum follows through the network is determined by the gating operations. Importantly, once determined by the gating, the network architecture is static and acts as a linear neural network. Thus, this static subnetwork is amenable to the same analysis as the deep linear networks above by utilising a layer-wise (each weight matrix connecting two vertices) singular value decomposition. The analysablity of the model is due to the linear nature of these "pathways" through the network which have no activation functions. As a result it is possible to explain their training dynamics as though they are linear neural networks (Saxe et al., 2014). The benefit of the layer-wise SVD dynamics approach is that the subnetworks being tracked do not compete or interfere as they are decoupled. As a result, all that is required to extend the deep linear network dynamics is to account for the frequency with which an edge (a set of weights) is used in the architecture graph. More frequently used edges will naturally learn quicker as a result. Thus, GDLNs are a class of neural network architectures composed of a number of linear neural networks which are selectively connected to different portions of the

input feature space, output feature space and only used for a subset of datapoints. The nonlinearity then comes in how these pathways are allocated which is controlled by a discrete gating variable.

Formally, let $\Gamma$ denote a directed graph with nodes $\mathbf{V}$ and edges $\mathbf{E}$. Each node $v \in \mathbf{V}$ represents a layer of neurons with activity $h_v \in \mathbb{R}^{|v|}$ where $|v| \in \mathbb{N}$ denotes the number of neurons in the layer. Each edge $e \in \mathbf{E}$ connects two nodes with a weight matrix $W_e$ of size $|t(e)| \times |s(e)|$ where $s, t : \mathbf{E} \to \mathbf{V}$ return the source and target nodes of the edge, respectively. It is also helpful to generalize this notion of edges to paths $p$, which are a sequence of edges $W_p$. Thus, paths also connect their source node $s(p)$ to their target node $t(p)$. We denote the portion of a path $p$ preceding a particular edge $e$ as $\bar{s}(p, e)$ and the portion of the path subsequent to the edge as $\bar{t}(p, e)$ such that $W_p = W_{\bar{t}(p,e)} W_e W_{\bar{s}(p,e)}$. We collect nodes with only outgoing edges into the set $\text{In}(\Gamma) \subset \mathbf{V}$ and call them input nodes. Similarly, output nodes only have incoming edges and are collected in the set $\text{Out}(\Gamma) \subset \mathbf{V}$. Activity is propagated through the network for a given datapoint which specifies values $x_v \in \mathbb{R}^{|v|}$ for all input nodes $v \in \text{In}(\Gamma)$. The activity of each subsequent layer is then given by $h_v = g_v \sum_{q \in \mathbf{E}:t(q)=v} g_q W_q h_{s(q)}$ where $g_v$ is a node gate and $g_q$ is an edge gate. Thus, the gating variables modulate the propagation of activity through the network by switching off entire nodes ($g_v$) and edges between nodes ($g_q$).

We train the GDLN to minimize the $L_2$ loss averaged over the full dataset of $N$ datapoints using gradient descent. The $i$-th datapoint then is a triple specified by the input $x^i$, output labels $y^i$ and specified gating structure $g^i$:

$$L(\{W\}) = \frac{1}{2N} \sum_{i=1}^{N} \sum_{v \in \text{Out}(\Gamma)} ||y_v^i - h_v^i||_2^2 \quad \text{where} \quad y_v \in \mathbb{R}^{|v|} \text{ for } v \in \text{Out}(\Gamma)$$

If we denote $\mathcal{P}(e)$ as the set of paths which pass through edge $e$ and $\mathcal{T}(v)$ as the set of paths terminating at node $v$ then the update step for a single weight matrix in the GDLN is:

$$\tau \frac{d}{dt} W_e = -\frac{\delta L(\{W\})}{\delta W_e} = \sum_{p \in \mathcal{P}(e)} W_{\bar{t}(p,e)}^T \left[ \Sigma^{yx}(p) - \sum_{j \in \mathcal{T}(t(p))} W_j \Sigma^x(j, p) \right] W_{\bar{s}(p,e)}^T \ \forall \, e \in \mathbf{E} \quad (8)$$

Thus, the update to a weight matrix $W_e$ is determined by the error at the end of a path which it contributes to:

$$\Sigma^{yx}(p) - \sum_{j \in \mathcal{T}(t(p))} W_j \Sigma^x(j, p)$$

summed over all paths it is a part of $p \in \mathcal{P}(e)$. Notably, all dataset statistics which direct learning are collected into the correlation matrices:

$$\Sigma^{yx}(p) = \frac{1}{N} \sum_{i=1}^{N} g_p^i y_{t(p)}^i x_{s(p)}^{iT}; \quad \Sigma^x(j, p) = \frac{1}{N} \sum_{i=1}^{N} g_j^i x_{s(j)}^i x_{s(p)}^{iT} g_p^i$$

*Crucially*, these dataset statistics now depend on the gating variables $g$, indicating that each path sees its own effective dataset determined by the network architecture. From this perspective, each path resembles the gradient flow of a deep linear network (Saxe et al., 2014; 2019) which have been shown to exhibit nonlinear learning dynamics observed in general deep neural networks (Baldi & Hornik, 1989; Fukumizu, 1998; Arora et al., 2018; Lampinen & Ganguli, 2019). The final step then is to use the change of variables employed by the linear network dynamics (Saxe et al., 2014; 2019). Assuming the effective dataset correlation matrices are mutually diagonalizable, we write them in terms of their singular value decomposition and the path weights in terms of the singular vectors:

$$\Sigma^{yx}(p) = U_{t(p)} S(p) V_{s(p)}^T; \quad \Sigma^x(j, p) = V_{s(j)} D(j, p) V_{s(p)}^T; \quad W_e(t) = R_{t(e)} B_e(t) R_{s(e)}^T \quad (9)$$

where $S(p)$ and $D(p)$ are diagonal matrices, $B_e(t)$ are the new dynamic variables, $R_v$ satisfies $R_v^T R_v = I$ for all $v \in \mathbf{V}$, and for input and output nodes $R_{s(p)} = V_{s(p)}$ and $R_{t(p)} = U_{t(p)}$, respectively. This change of variables removes competitive interactions between singular value

modes along a path such that the dynamics of the overall network is described by summing several "1D networks" (one for each singular value) resulting in the "neural race reduction" (Saxe et al., 2022):

$$\tau \frac{d}{dt} B_e = \sum_{p \in \mathcal{P}(e)} B_{p \backslash e} \left[ S(p) - \sum_{j \in \mathcal{T}(t(p))} B_j D(j, p) \right] \ \forall \, e \in \mathbf{E} \tag{10}$$

This reduction demonstrates that the learning dynamics depend both on the input-output correlations of the effective path datasets and the number of paths to which an edge contributes. Ultimately, the paths which learn the fastest from both pressures will win the neural race.

## B   An Algorithm for Identifying Gating Patterns

In this work, our findings on ReLU networks did not rely on the manner that the ReLN (specifically the appropriate gating patterns) is obtained. This is also why it was necessary to prove the uniqueness of the ReLN, for example Proposition 4.2 demonstrates that there is no other ReLN which matches the ReLU loss trajectory perfectly in this case. Thus, we could be certain of our findings regardless of where the gates came from. However, since a contribution is the general connection between the ReLU and GDLN, it is helpful to provide a simple algorithm which can identify the ReLN in the space of GDLN architectures. We note that it is always possible to find a GDLN which imitates the ReLU network, as we discuss in Section 3.

Our algorithm for identifying ReLNs is shown in Algorithm 1 and is based on a simple k-Means clustering. The primary idea is that we sample the hidden layer representations from the ReLU network at various stages throughout training. This makes it easier to detect certain structures which emerge in the gating patterns - such as the common pathway emerging early in training. We then run the training of the network multiple times to mitigate the impact of symmetries emerging which do not impact the functional mapping of the network, such as permutations and neuron splitting (Simsek et al., 2021; Martinelli et al., 2023). Each sampling from the hidden representation will have shape $(H, N)$ where $H$ is the number of hidden neurons and $N$ is the number of datapoints. We stack these samples across training runs and timesteps (vertically). Thus, if $C$ samples are taken then the ultimate set of datapoints to be clustered will be of shape $(C * H, N)$. Conceptually, this is $C * H$ hidden neurons all performing some gating over the dataset. We cluster these hidden neurons (the $C * H$ rows) with k-Means which results in $k$ clusters with $N$-dimensional centroids. As is typical of k-Means, $k$ is a hyper-parameters which we choose. These centroids provide the gating pattern for one *type* of hidden neuron and will become a gating pattern in the GDLN. Conceptually, if enough neurons are of the same type, they form a modules and it is these implicit modules we aim to make explicit in the ReLN.

---

**Algorithm 1** *A preliminary algorithm for finding a ReLN.* This follows a simple K-means clustering algorithm, but with samples taken throughout training such that it is easier to identify pathways through the network as they emerge.

---

**Require:** $num\_trainings > 0, num\_epochs > 0, \sigma > 0, (X \in \mathbb{R}^{d \times N}, Y \in \mathbb{R}^{p \times N})$ (the dataset), $H \in \mathbb{Z}, K \in \mathbb{Z}$
**Ensure:** $\sigma < \epsilon$ for sufficiently small $\epsilon \in \mathbb{R}$
  **for** $i$ in $num\_trainings$ **do**
    $\bar{W}_0 \in \mathbb{R}^{H \times d} \sim \mathcal{N}(0, \sigma), \bar{W}_1 \in \mathbb{R}^{p \times H} \sim \mathcal{N}(0, \sigma)$
    **for** $j$ in $num\_epochs$ **do**
      $\{\bar{W}_0, \bar{W}_1\} \leftarrow$ gradient\_descent($\{\bar{W}_0, \bar{W}_1\}, X$)    ▷ Apply gradient descent update step
      **if** j mod 100 = 0 **then**  ▷ Sample at different times to find different structure as it emerges
        sample = maximum($\bar{W}_0 X, 0$)   ▷ Sample latent representations with ReLU activation
        sample\_binary = $step$(sample) ▷ Threshold the sample to indicate if a neuron is active
        samples = vstack(samples,sample\_binary)    ▷ Stack binary latent representations
      **end if**               ▷ Each sample appended vertically appears like a new neuron
    **end for**
  **end for**
  centroids = K-means(samples,$K$)
  **return** centroids

---

The results of using this algorithm to identify a ReLN for the 3-context dataset with different structure between datasets (see Section 4) is shown in Figure 8. Each row in this figure corresponds to a cluster and the columns correspond to the datapoints in the dataset. For example, the green row in the 2-Clusters set reflects the dominant gating pattern of one cluster of hidden neurons across the whole dataset. In this case, the hidden neurons allocated to this green cluster were typically active in context 2 and 3 but inactive in the first context (the first 8 columns have no activity which corresponds to the datapoints in the first context). We show the clustering when $k = 2$, $k = 3$ and $k = 4$. We note that all of the clusterings find a common pathway, likely due to its emergence early on in training. From there they begin to identity mixed selective modules (which are active in two contexts). However, it is worth noting the effect of averaging on these clusters. This is easiest to see in the 3 Clusters case where the red cluster which is active for all contexts is also trying to capture the cluster for the hidden neurons active in context 1 and 3. Thus, the middle portion of that cluster is dulled compared to the others. In such cases it is difficult to interpret what exactly the gating pattern should be and is a sign that more clusters are needed. However, this indicates a limitation of this approach - it relies on the clusters to have a consistent gating pattern for each datapoint throughout training. We do not limit the GDLN to such a condition. We note that when using 4-clusters the exact gating pattern used in Section 4 emerges.

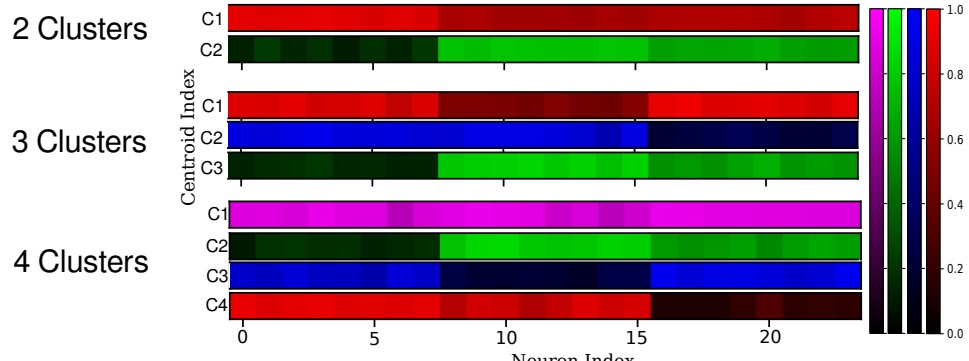

Figure 8: **Clustering to find ReLNs**: Clusters found with varying number of centroids for the dataset in Section 4. Note the inconsistency in gating patterns arising from having too few clusters. Consequently, centroids have neuron activity between $0.0$ and $1.0$ indicating that the gating is not consistent across data points.

Since the algorithm is based on clustering, it is still necessary to pick the number of centroids appropriately. This has a clear effect on the performance of the model as the number of clusters will dictate the number of unique gating patterns our GDLN can use to imitate the ReLU network. We can, however make this less qualitative by using the typical approach of finding an elbow in a plot of the number of clusters compared to a performance metric. In this case the performance metric is very rich and can be the mean-squared error between the ReLU network and the GDLN with the identified gating patterns. This can give a very precise measure of whether the number of clusters was correct. We show this elbow in Figure 9 and see that the 4-Cluster model is the appropriate choice for the elbow, as expected.

The algorithm presented here can be seen as a very basic form of meta-learning for a GDLN, in the sense that the GDLN is being trained for a meta-task of imitating the ReLU network, while learning to minimize its own loss. However, the space of GDLNs is also larger than the space of ReLU networks. For example, ReLU networks are only able to implement gating patterns as a function of their input, while GDLNs are capable of gating based on rules, external patterns or previous datapoints. Here we are concerned with GDLNs which are restricted to gating patterns implementable by ReLU networks. However, many works have demonstrated a benefit from imposed architecturally modular networks which specialize towards solving particular and interpretable sub-problems within a given task (Andreas et al., 2016; Hu et al., 2017; Vani et al., 2021). Indeed, one proposed direction to obtain modular networks is through meta-learning (Lake, 2019; Lake & Baroni, 2023). Mixed-selectivity, and the closely related concept of polysemanticity (neurons being activated by multiple semantic unit or features) (Olah et al., 2017a; Lecomte et al., 2024) have typically been thought to be unrelated so such modularity and the systematic generalization (Ruis & Lake, 2022). However, as we have shown in this work ReLU networks are able to form interpretable modules across contexts which

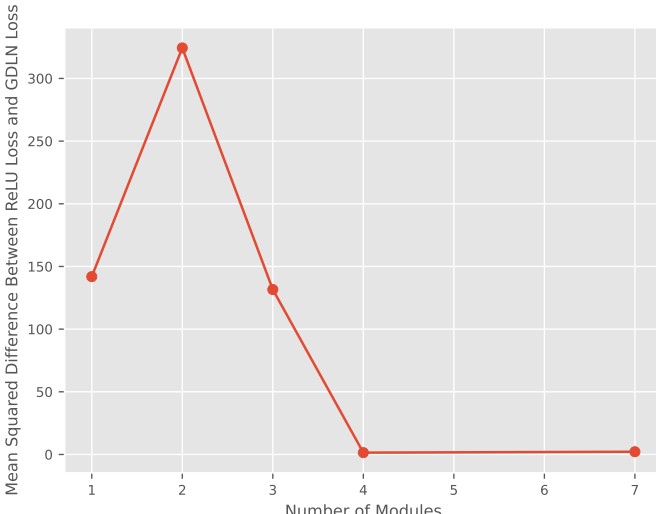

Figure 9: **Clustering to find ReLNs**: Finding the elbow to determine the minimal number of clusters (linear modules in the ReLN).

are mixed selective and form coupled modules. Thus, a more advanced meta-learning strategy for GDLNs, than the approach of this section, which learns the gates based on an entirely different rule or set of features may allow a GDLN to perform computations unavailable to the ReLU network and by extension a ReLN. This appears to be a promising future direction.

## C    LEARNING DYNAMICS DURING TRANSITION TO NONLINEAR SEPARABILITY

Deep neural networks can learn representations suitable for solving nonlinear tasks. Here we investigate their learning dynamics on a simple class of tasks. We will qualitatively understand the dynamics of training a single hidden layer ReLU network by reducing it a gated deep linear network.

**Dataset.** The canonical example of a nonlinear problem is the XoR problem. We consider a slight generalization: an XoR problem in the first two input dimensions, but a linearly separable problem in the third. Our dataset $X \in R^{3 \times 4}, y \in R^{1 \times 4}$ consists of four data points in three dimensions,

$$X \quad = \quad \begin{bmatrix} -1 & 1 & -1 & 1 \\ -1 & -1 & 1 & 1 \\ -\Delta & \Delta & \Delta & -\Delta \end{bmatrix} \tag{11}$$

$$y \quad = \quad \begin{bmatrix} -1 & 1 & 1 & -1 \end{bmatrix} \tag{12}$$

where $\Delta \geq 0$ is a parameter that controls the linear separability of the points along the third dimension.

Our goal will be to understand how a ReLU network transitions from a regime in which the data points are clearly linearly separable ($\Delta >> 1$) to the regime in which they are not ($\Delta = 0$, the classical XoR task).

**GDLN with linear gating structure.** We first solve the dynamics of the GDLN depicted in Figure 2b of the main text. This network contains two pathways, one gated on for the positive examples and one gated on for the negative examples.

The effective dataset statistics for the positive examples, which we denote as $\Sigma^{yx}(p)$, is therefore

$$\Sigma^{yx}(p) \quad = \quad \frac{1}{4} \begin{bmatrix} 1 & 1 \end{bmatrix} \begin{bmatrix} 1 & -1 \\ -1 & 1 \\ \Delta & \Delta \end{bmatrix}^{\top} \tag{13}$$

$$= \quad \begin{bmatrix} 0 & 0 & \frac{\Delta}{2} \end{bmatrix} \tag{14}$$

$$\Sigma^x(p,p) = \frac{1}{4} \begin{bmatrix} 1 & -1 \\ -1 & 1 \\ \Delta & \Delta \end{bmatrix} \begin{bmatrix} 1 & -1 \\ -1 & 1 \\ \Delta & \Delta \end{bmatrix}^\top \tag{15}$$

$$= \frac{1}{2} \begin{bmatrix} 1 & -1 & 0 \\ -1 & 1 & 0 \\ 0 & 0 & \Delta^2 \end{bmatrix}. \tag{16}$$

By symmetry, the loss dynamics on the negative examples (driven by effective statistics $\Sigma^{yx}(n)$) will be identical to the positive examples. Because only one pathway is active on any example, they develop independently.

To solve the dynamics, we leverage analytical solutions for deep linear networks describing each pathway. The dynamics of a deep linear network depend on the singular value decomposition of the effective input-output correlations $\Sigma^{yx}(p) = USV^\top$ and input correlations $\Sigma^x(p,p) = VDV^\top$.

The singular value decomposition is
$$\Sigma^{yx}(p) = \begin{bmatrix} 0 & 0 & \frac{\Delta}{2} \end{bmatrix} = [1] \begin{bmatrix} \frac{\Delta}{2} \end{bmatrix} \begin{bmatrix} 0 & 0 & 1 \end{bmatrix} \tag{17}$$
so the singular value is $s = \Delta/2$. The input variance in this singular vector direction is $d = \Delta^2/2$. From the known time course of deep linear networks, the effective singular value in this pathway will be

$$a(t) = \frac{s/d}{1 - (1 - \frac{s}{da_0})e^{-2st/\tau}} \tag{18}$$

$$= \frac{1/\Delta}{1 - (1 - \frac{1}{\Delta a_0})e^{-\Delta t/\tau}}. \tag{19}$$

The loss from this pathway $\mathcal{L}_p$ therefore is
$$\mathcal{L}_p(t) = \mathrm{Tr}\Sigma^y(p)/2 - sa(t) + 1/2da(t)^2 \tag{20}$$
$$= 1/4 - \Delta a(t)/2 + \Delta^2 a(t)^2/4. \tag{21}$$
By symmetry, the contribution of the other pathway for the negative examples is identical. Hence the total loss is twice this single-pathway loss, yielding
$$\mathcal{L}(t) = 1/2 - \Delta a(t) + \Delta^2 a(t)^2/2, \tag{22}$$
the expression plotted in Figure 2e (blue) of the main text.

**GDLN with XoR gating structure.** If instead we have a GDLN with four pathways each gated on for exactly one input sample, then the effective dataset only contains one input sample. Taking the pathway gated on for the first example, for instance, the singular value of $\Sigma^{yx}(1) = \frac{1}{4}[1 \quad 1 \quad \Delta]$ is $s = \sqrt{\frac{1}{8} + \frac{\Delta^2}{16}}$. The associated input variance is $d = \frac{1}{8} + \frac{\Delta^2}{16}$. Substituting these values into Eqn. 19 yields the effective singular value trajectory, and similar steps yield the total loss
$$\mathcal{L}(t) = 1/2 - 4sa(t) + 2da(t)^2, \tag{23}$$
the equation plotted in Figure 2e (orange) of the main text.

By symmetry the remaining three datapoints have identical loss dynamics. Because all pathways concern disjoint examples, they evolve independently.

We now ask for what range of $\Delta$ will the linear separability GDLN learn faster than the XoR GDLN. The critical $\Delta$ separating these regimes is the one where both GDLNs learn equally quickly. The learning speed of deep linear networks is of order $O(1/s)$ where $s$ is the singular value. The crossover point is therefore

$$\Delta/2 = \sqrt{\frac{1}{8} + \frac{\Delta^2}{16}} \tag{24}$$

$$4\Delta^2 = 2 + \Delta^2 \tag{25}$$

$$\Delta = \sqrt{2/3}. \tag{26}$$

By testing points, we see that for $0 \leq \Delta < \sqrt{2/3}$ the XoR GDLN is faster while for $\Delta > \sqrt{2/3}$ the linear separability GDLN is faster. Hence a ReLU network does not always exploit linear separability. The linear margin is $2\Delta$ but for small enough margins, the race will be won by the XoR GDLN, implying a nonlinear decision boundary.

# D  PROOF OF LEMMA 4.1

**Lemma 4.1:** *On the input X (as defined in Section 4) and given the definition of GDLNs and ReLU Networks, for any gating pattern implementable by the ReLU network there is $G(x_c)$ such that $G(x_c)_j = step(\overline{h}_j)$ or $G(x)_j = \mathbb{1}(x) \; \forall \; j \in [0, H]$ where $\overline{h}$ is the pre-activations of the ReLU network's hidden layer with H neurons, j is the index of a hidden neuron in either network and $x_c$ is the portion of the data point x with contextual features.*

**Proof:** We consider which functions are implementable by a ReLU network given that it is restricted to using the same weights for both gating and association. Without loss of generality we consider the simplest instantiation of two objects with two contexts. Thus, there are four datapoints, specifically $x_1 = [1, 0, 1, 0]$, $x_2 = [0, 1, 1, 0]$, $x_3 = [1, 0, 0, 1]$ and $x_4 = [0, 1, 0, 1]$.

First, we consider if combined gating strategies can be implemented. By a combined gating strategy what we mean is that the gate is determined by both the object and context feature in combination. Thus there must be at least one object which is gated differently for the same context as another object (or else the gating could just be done based on context). Similarly there must be a context where an objects gating changes (or else the gating could be done based on object). In other words, there is no consistent gating for object or context in isolation across the whole dataset. Assume a combined gating exists. This means there is at least one hidden neuron which is on for $x_2$ and $x_3$ but off for $x_1$ and $x_4$. Thus, the network must have sufficiently negative weights connecting to this neuron such that its pre-activation is negative on $x_1$. Let these weights be $[a, b, c, d]$. Then $a + c <= 0$ and $b + d <= 0$. This means $a <= -c$ and $b <= -d$. Conversely, $b + c > 0$ and $a + d > 0$. Thus, $b > -c$ and $a > -d$. Taken together $a <= -c$ and $b > -c$ means that $a < b$. Additionally, $b <= -d$ and $a > -d$ means that $b < a$ which is a contradiction. Thus, no such hidden neuron with a combined gating can exist. This means that the network must gate consistently for either context or object features.

The gating strategy based only on items can be rules out as the mapping from context to output is nonlinear by construction. Thus, regardless of the number of latent neurons there is no linear mapping from contexts to output alone which will achieve no error. There is an edge case to this proof which we cannot rule out, that of allocating a single neuron for each data point. Here we cannot obtain a contradiction as there is no other data points to contradict with. This is also the typical strategy employed by ReLU networks on XoR tasks such as this one. We will cover this case in Proposition 4.2. Thus, the only remaining gating strategies implementable by a one-layer ReLU network on the dataset described in Section 4) is to gate consistently for the context variables. This leaves the common pathway, a second of contextual pathways active for two contexts, and finally contextual pathways active for a single context.

# E  PROOF OF PROPOSITION 4.2

**Proposition 4.2:** *For input X defined in Section 4) there is a unique $G^*(X)$ (up to symmetries such as permutations) such that $L_{GDLN}(W(t), G^*(X)) = L_{ReLU}(W(t)) \; \forall \; t \in \mathbb{R}^+$ and for alternative gating strategies $G'(X)$ then $L_{GDLN}(W(t), G^*(X)) < L_{GDLN}(W(t), G'(X))$*

**Proof:** To begin we prove a generalization of the Cauchy Interlacing Theorem for non-square matrices which closely follows the strategy of Thompson (1972). The aim of this is to show that if an input or output feature is removed from a pathway's effective dataset that the first singular value will decrease. Since the singular values determine the loss trajectory of the network, this will then mean that the loss curve will no longer fit the ReLU network. Thus:

For any matrix $M$ lets denote the submatrix of rows $i_1, ..., i_p$ and columns $j_1, ..., j_q$ as $M[i_1, ..., i_p | j_1, ..., j_q]$. To ease notation let $\mu = \{i_1, ..., i_p\}$ and $\nu = \{j_1, ..., j_q\}$. Then $B = \Sigma_{yx}[\mu|\nu]$ since it is a submatrix of $\Sigma_{yx}$. If $U = \mathbb{I}_m$ and $V = \mathbb{I}_n$ (the identity matrices of $m$ and $n$ dimensions) then $B$ can also be written as $B = U[\mu|1, ..., m]\Sigma_{yx}V[1, .., n|\nu]$. Thus:

$$BB^T = U[\mu|1, ..., m]\Sigma_{yx}V[1, .., n|\nu]V^T[\nu|1, .., n]\Sigma_{yx}^T U^T[1, ..., m|\mu]$$

$$= U[\mu|1, ..., m]AA^T U^T[1, ..., m|\mu]$$

where $A = \Sigma_{yx}V[1, .., n|\nu] \in \mathbb{R}^{m \times q}$ with ordered, non-zero singular values of $\alpha_1, \alpha_2, ..., \alpha_{\min(m,q)}$. Thus $BB^T$ is a principal $p$-square submatrix of the $m$-square symmetric matrix $UAA^T U^T$. Since

$BB^T$ is guaranteed to be symmetric we know it will be diagonalizable. Thus, there exists an orthonormal basis of eigenvectors $\{b_1, ..., b_{\min(p,q)}\}$ corresponding to eigenvalues $\{\beta_1^2, ..., \beta_{\min(p,q)}^2\}$. We define $G_j = \text{span}[b_1, ..., b_j]$ (the span of the first $j$ eigenvectors of $BB^T$) for $j \leq \min(p, q)$ and $S_j = \text{span}[a_j, ..., a_{\min(m,q)}]$ (the span of the last $\min(m, q) - j + 1$ eigenvectors of $AA^T$). We also define the subspace:

$$H_j = \left\{ U^T[1, ..., m|\mu]g, g \in G_j \right\}$$

There exits a unit length vector $\tilde{z} = U^T[1, ..., m|\mu]z$ for $z \in G_j$ which lies in $H_j \cap S_j$, as if there is not then the dimension of $H_j \cap S_j$ would be $j + \min(m, q) - j + 1$ which is impossible in $\mathbb{R}^{\min(m,q)}$. Additionally:

$$z = \sum_{i=1}^{j} r_i b_i \qquad \text{and} \qquad \langle z, z \rangle = z^T z = \sum_{i=1}^{j} r_i^2 = 1$$

(since the eigenvectors of symmetric matrices are orthogonal). We begin by considering the Rayleigh–Ritz quotient for $BB^T$ and a unit length vector $v$:

$$R_{BB^T}(v) = \frac{\langle BB^T v, v \rangle}{\langle v, v \rangle} = v^T BB^T v = \sum_{i=1}^{\min(p,q)} \beta_i^2 r_i^2$$

Here $r_i^2$ acts as a weighting on each eigenvalue for how much it contributes to the sum. Thus to minimize the quotient we place all of the weighting on the lowest eigenvalue $\beta_j^2$ by picking $v$ such that $r_i = 0$ for $i \in \{1, ..., j-1\}$ and $r_j = 1$. This result is known as the Min-Max Theorem. A similar result can be shown where the $j$-th eigenvalue also results from maximizing the quotient over the span of the bottom eigenvector beginning with the $j$-th eigenvector. Thus:

$$\beta_j^2 = \min_{v \in G_j; ||v||=1} R_{BB^T}(v) \text{ by the Min-Max Theorem}$$

$$= \min_{v \in G_j; ||v||=1} \langle BB^T v, v \rangle$$

$$\leq \langle BB^T z, z \rangle \text{ since } z \in G_j \text{ does not necessarily minimize the quotient}$$

$$= \langle U[\mu|1, ..., m]AA^T U^T[1, ..., m|\mu]z, z \rangle$$

$$= \langle AA^T \left( U^T[1, ..., m|\mu]z \right), U^T[1, ..., m|\mu]z \rangle \text{ by the linearity of the inner product}$$

$$= \langle AA^T \tilde{z}, \tilde{z} \rangle$$

$$\leq \max_{\tilde{v} \in S_j; ||v||=1} \langle AA^T \tilde{v}, \tilde{v} \rangle \text{ since } \tilde{z} \in S_j \text{ does not necessarily maximize the quotient like } \tilde{v}$$

$$= \max_{\tilde{v} \in S_j; ||v||=1} R_{AA^T}(\tilde{v})$$

$$= \alpha_j^2 \text{ by the Min-Max Theorem}$$

Thus $\beta_j \leq \alpha_j$. Similarly, $A^T A = V^T[\nu|1, .., n]\Sigma_{yx}^T \Sigma_{yx} V[1, .., n|\nu]$ with eigenvalues of $\alpha_1^2, \alpha_2^2, ..., \alpha_{\min(m,q)}^2$. Thus, $A^T A$ is a principal $q$-square submatrix of the $n$-square symmetric matrix $V^T \Sigma_{yx}^T \Sigma_{yx} V$. Since $A^T A$ is guaranteed to be symmetric we know it will be diagonalizable. Thus, there exists an orthonormal basis of eigenvectors $\{a_1, ..., a_{\min(m,q)}\}$ corresponding to eigenvalues $\{\alpha_1^2, ..., \alpha_{\min(m,q)}^2\}$. We overload the notation and define $G_j = \text{span}[a_1, ..., a_j]$ (the span of the first $j$ eigenvectors of $A^T A$) for $j \leq \min(m, q)$ and $S_j = \text{span}[\sigma_j, ..., \sigma_{\min(m,n)}]$ (the span of the last $\min(m, n) - j + 1$ eigenvectors of $\Sigma_{yx}^T \Sigma_{yx}$). We also define the subspace:

$$H_j = \{V[1, .., n|\nu]g, g \in G_j\}$$

There exits a unit length vector $\tilde{z} = V[1, .., n|\nu]z$ for $z \in G_j$ which lies in $H_j \cap S_j$, as if there is not then the dimension of $H_j \cap S_j$ would be $j + \min(m, n) - j + 1$ which is impossible in $\mathbb{R}^{\min(m,n)}$. Additionally:

$$z = \sum_{i=1}^{j} r_i a_i \qquad \text{and} \qquad \langle z, z \rangle = z^T z = \sum_{i=1}^{j} r_i^2 = 1$$

(since the eigenvectors of symmetric matrices are orthogonal). We begin by considering the Rayleigh–Ritz quotient for $A^T A$ and a unit length vector $v$:

$$R_{A^T A}(v) = \frac{\langle A^T A v, v \rangle}{\langle v, v \rangle} = v^T A^T A v = \sum_{i=1}^{\min(m,q)} \alpha_i^2 r_i^2$$

Here $r_i^2$ acts as a weighting on each eigenvalue for how much it contributes to the sum. Thus to minimize the quotient we place all of the weighting on the lowest eigenvalue $\alpha_j^2$ by picking $v$ such that $r_i = 0$ for $i \in \{1, ..., j-1\}$ and $r_j = 1$:

$$\alpha_j^2 = \min_{v \in G_j; ||v||=1} R_{A^T A}(v) \text{ by the Min-Max Theorem}$$

$$= \min_{v \in G_j; ||v||=1} \langle A^T A v, v \rangle$$

$$\leq \langle A^T A z, z \rangle \text{ since } z \in G_j \text{ does not necessarily minimize the quotient}$$

$$= \langle V^T[\nu|1,..,n]\Sigma_{yx}^T \Sigma_{yx} V[1,..,n|\nu] z, z \rangle$$

$$= \langle \Sigma_{yx}^T \Sigma_{yx} \left( V[1,..,n|\nu] z \right), V[1,..,n|\nu] z \rangle \text{ by the linearity of the inner product}$$

$$= \langle \Sigma_{yx}^T \Sigma_{yx} \tilde{z}, \tilde{z} \rangle$$

$$\leq \max_{\tilde{v} \in S_j; ||v||=1} \langle \Sigma_{yx}^T \Sigma_{yx} \tilde{v}, \tilde{v} \rangle \text{ since } \tilde{z} \in S_j \text{ does not necessarily maximize the quotient like } \tilde{v}$$

$$= \max_{\tilde{v} \in S_j; ||v||=1} R_{\Sigma_{yx}^T \Sigma_{yx}}(\tilde{v})$$

$$= \sigma_j^2 \text{ by the Min-Max Theorem}$$

Thus $\alpha_j \leq \sigma_j$. Putting both inequalities together we obtain the result: $\beta_j \leq \alpha_j \leq \sigma_j$ with the particularly important case of $\beta_1 \leq \sigma_1$. This means that a shared pathway will always be learned faster than a pathway which considers only a subset of the input features or output labels (which input features is determined by $\mu$ and which output labels is determined by $\nu$).

Next we consider whether a different gating strategy exists which gates along datapoints instead of features. We note that the output of the common pathway is element-wise positive for all datapoints and all input features are positive. Consequently, let $\Sigma^{yx} = YX^T$ be the correlation matrix of this dataset. If we factor out $Y$ as $Y = \hat{Y} + Z$ where $\hat{Y}$ is $Y$ with a datapoint removed (its column set to 0) and $Z$ is the matrix of zeros everywhere except for the column removed from $Y$ where it is equal to $Y$. Then $\Sigma^{yx} = YX^T = (\hat{Y} + Z)X^T = \hat{Y}X^T + ZX^T$. Now let $\hat{v}$ be the eigenvector corresponding to the largest eigenvalue of $\hat{Y}X^T X \hat{Y}$ and v be the top eigenvector for the full dataset $\Sigma^{yx}\Sigma^{yx^T}$. Then $\Sigma^{yx}\Sigma^{yx^T} v \geq \Sigma^{yx}\Sigma^{yx^T}\hat{v} = (\hat{Y}X^T + ZX^T)(\hat{Y}X^T + ZX^T)^T \hat{v} = \hat{Y}X^T X \hat{Y}\hat{v} + \hat{Y}X^T X Z^T \hat{v} + ZX^T X \hat{Y}^T \hat{v} + ZX^T X Z^T \hat{v} \geq \hat{Y}X^T X \hat{Y}\hat{v}$ since $v$ will be in the all positive orthant where the entire dataset lies. Thus, the common pathway can only train faster with the addition of more datapoints. Thus, the common pathway is the fastest possible subnetwork for the given dataset, and any other module will train slower (there is an edge case here where the added datapoint is orthogonal to all previous datapoints, in which case $XZ^T = 0$, however no such datapoint exists in this dataset).

We can now rule out the edge case from Lemma 4.1. Once the common pathway has finished learning there is no correlation between the context features and labels. As a consequence in a case when a single datapoint is allocated to a single module all activity would need to be learned by the item feature. In our dataset, removing all other items in a context would then be the same as removing their individual input features. From the Cauchy Interlacing Theorem, we know this can only slow learning.

Finally, we consider whether alternatives to the context sensitive pathways exists. We note that from Lemma 4.1 with the addition of a common pathway, there are only two available options to the remaining pathways, as these pathways can only gate consistently using context features. Thus, either the modules must be active for one or for two contexts. In this case we can easily simulate this to determine which is quickest and we see in Figure 4 that the architecture which uses modules that are active for two contexts wins the race. Indeed even if all six possible modules are trained simultaneously, the pathways active for a single context remain inactive. Thus, the pathways active for two contexts have a unique loss trajectory and when paired with the common pathway have a unique and fastest loss trajectory. Thus, the ReLU network uncovers the neural race winner.

# F  GDLN DYNAMICS WITH THREE CONTEXTS

We now derive the dynamics when there are three context specific portions of the output space. Specifically, we have a dataset $D = (\mathcal{I}, C, Y)$ with object, context and output feature matrices $\mathcal{I} \in \mathbb{R}^{N \times N}$, $C \in \mathbb{R}^{3 \times N}$ and $Y \in \mathbb{R}^{2N-1 \times N}$ respectively. Here $N$ denotes the number of datapoints. We concatenate the object and context features vertically to form the input matrix: $X \in \mathbb{R}^{N+3 \times N}$. Every object (depicted by a one-hot input vector in $\mathcal{I}$) can be encountered in every context (depicted by another one-hot vector in $C$). It is convenient to partition the input matrix into three blocks with the same one-hot context vector: $X = [X_1, X_2, X_3]$ where $X_i \in \mathbb{R}^{N+3 \times N/3}$. Similar partitions follow for $\mathcal{I}$ and $C$ separately. Both models are told which of the objects to produce the features for as well as in which context the object is encountered in. The models must then produce the correct output features, some of which are appropriate for all contexts and others only in one specific context.

Our ReLN trained to imitate the ReLU network loss trajectory and dynamics is composed of four pathways. All pathways have a linear layer of 100 neurons resulting in the same number of total hidden neurons as the ReLU network. The first pathway receives all objects and contexts and maps to the full output matrix. We call this the common pathway and denote it by $*$, with weight and output mapping: $W_* = \{W_*^1, W_*^2\}$ and $\hat{y}^* = W_*^2 W_*^1 X$. Thus, to model this pathway's loss trajectory we apply the linear dynamics equations of Saxe et al. (2014) to the input-output correlation: $\Sigma^* = YX^T = U_* S_* V_*^T$. The common pathway's dynamics then follow $W_*^2 W_*^1 = U_* A_*(t) V_*^T$ and will have the mapping $\hat{Y}^*(t) = U_* A(t)_* V_*^T X$ at all points in time. It is assumed that the network aligns to the dataset singular vectors immediately with $U_*$ and $V_*^T$. The dynamics of the network can then be obtained through the decoupled singular value dynamics of:

$$\pi_\alpha(t) = \frac{S_\alpha / D_\alpha}{1 - (1 - \frac{S_\alpha}{D_\alpha \pi_0}) \exp(\frac{-2S_\alpha}{\tau} t)}$$

where $S_\alpha$ is the $\alpha$-th mode of the input-output correlation singular value matrix $S$, $D_\alpha$ is similarly the $\alpha - th$ mode of the input correlation matrix $(XX^T)$ and $\pi_0$ is the initial state of the network singular values $\pi_\alpha$. Thus, $\pi_\alpha$ begins with a value of $\pi_0$ and increases until its convergence point at $S_\alpha / D_\alpha$. We can then define the residual of this model as $Y^* = Y - \hat{Y}^*(\infty)$.

The three remaining (context specific) pathways are then trained on this residual. The context specific pathways are each gated on for two context settings and connect to all output features. In addition, while these pathways are used in two contexts they are not shown the input context as a feature. The result is that these pathways can only use the input features $\mathcal{I}_i$ in their mapping when they are gated on. Since these pathways are trained at once and have dynamics which depend on the other pathways we now use the GDLN framework (Saxe et al., 2022) to determine their dynamics (we review this setup in Section 2 in the main text). The learning rule for a weight matrix in a GDLN follows the update equation (we use the brackets $\langle \cdot \rangle$ to denote averaging as typically used in statistical physics to simplify notation):

$$\tau \frac{d}{dt} W_e = -\frac{\delta L(\{W\})}{\delta W_e} \forall e \in E$$

$$= \left\langle \sum_{p \in P(e)} g_p W_{\bar{t}(p,e)}^T \left[ y_{t(p)} x_{s(p)}^T - h_{t(p)} x_{s(p)}^T \right] W_{\bar{s}(p,e)}^T \right\rangle_{y,x,g}$$

We note that for our architecture there is only one pathway that each edge is involved in. Thus we may drop the summation over pathways and denote the one relevant pathway with $p$. We also apply the usual linearity of the expectation:

$$= W_{\bar{t}(p,e)}^T \left[ \left\langle g_p y_{t(p)} x_{s(p)}^T \right\rangle_{y,x,g} - \left\langle g_p h_{t(p)} x_{s(p)}^T \right\rangle_{y,x,g} \right] W_{\bar{s}(p,e)}^T$$

The next step is to determine the output of the network which impacts pathway $p$. Specifically we substitute in for $h_{t(p)}$ the sum of all pathways mapping onto $t(p)$ - the terminal point for pathway $p$:

$$h_{t(p)} = \sum_{j \in \mathcal{T}(t(p))} g_j W_j x_{s(j)}$$

Here $\mathcal{T}(t(p))$ is the set of all pathways leading to the same terminal node as pathway $p$. Thus $j$ denotes each of these pathways with corresponding mapping $W_j$ (note $W_j$ here is the product of all matrices along pathway $j$). Finally $x_{(s(j))}$ is the input to pathway $j$ since $s(j)$ denotes the source of $j$. Substituting this into the dynamics equation we obtain:

$$= W_{\bar{t}(p,e)}^T \left[ \left\langle g_p y_{t(p)} x_{s(p)}^T \right\rangle_{y,x,g} - \left\langle g_p \sum_{j \in \mathcal{T}(t(p))} g_j W_j x_{s(j)} x_{s(p)}^T \right\rangle_{y,x,g} \right] W_{\bar{s}(p,e)}^T$$

$$= W_{\bar{t}(p,e)}^T \left[ \left\langle g_p y_{t(p)} x_{s(p)}^T \right\rangle_{y,x,g} - \sum_{j \in \mathcal{T}(t(p))} W_j \left\langle g_j x_{s(j)} x_{s(p)}^T g_p \right\rangle_{y,x,g} \right] W_{\bar{s}(p,e)}^T$$

$$= W_{\bar{t}(p,e)}^T \left[ \Sigma^{yx}(p) - \sum_{j \in \mathcal{T}(t(p))} W_j \Sigma^x(j,p) \right] W_{\bar{s}(p,e)}^T$$

Thus, the dynamics for one layer of weights ($W_e$) in a single pathway ($p$) depends on the input-output correlation for that pathway ($\Sigma^{yx}(p)$), the input correlation for the pathway itself ($\Sigma^x(p,p)$) and the input correlation for the pathway with the input to all other pathways sharing the terminal point ($\Sigma^x(j,p); j \neq p$). Thus we rewrite the dynamics in terms of these three kinds of correlations separately:

$$= W_{\bar{t}(p,e)}^T \left[ \Sigma^{yx}(p) - W_p \Sigma^x(p,p) - \sum_{j \neq p \in \mathcal{T}(t(p))} W_j \Sigma^x(j,p) \right] W_{\bar{s}(p,e)}^T$$

We note then that for the particular setting we are working in $\Sigma^x(j,p) = \frac{1}{2}\Sigma^x(p,p)$. Thus, we denote $\Sigma^x = \Sigma^x(p,p)$ and substitute this relationship into the dynamics above. Now the input correlation terms do not depend on the alternative pathways taken to the terminal points.

$$= W_{\bar{t}(p,e)}^T \left[ \Sigma^{yx}(p) - W_p \Sigma^x - \sum_{j \neq p \in \mathcal{T}(t(p))} W_j \frac{1}{2}\Sigma^x \right] W_{\bar{s}(p,e)}^T$$

$$= W_{\bar{t}(p,e)}^T \left[ \Sigma^{yx}(p) - W_p \Sigma^x - \left( \frac{1}{2} \sum_{j \neq p \in \mathcal{T}(t(p))} W_j \right) \Sigma^x \right] W_{\bar{s}(p,e)}^T$$

$$\tau \frac{d}{dt} W_e = W_{\bar{t}(p,e)}^T \left[ \Sigma^{yx}(p) - \left( W_p + \frac{1}{2} \sum_{j \neq p \in \mathcal{T}(t(p))} W_j \right) \Sigma^x \right] W_{\bar{s}(p,e)}^T$$

We now apply the change of variables to obtain the dynamics reduction:

$$\tau \frac{d}{dt} \left( R_{t(e)} B_e R_{s(e)^T} \right) = \left( U_{t(p)} B_{\bar{t}(p,e)} R_{t(e)}^T \right)^T \left[ U_{t(p)} S(p) V_{s(p)}^T - \right.$$

$$\left. \left( U_{t(p)} B_p V_{s(p)}^T + \frac{1}{2} \sum_{j \neq p \in \mathcal{T}(t(p))} U_{t(j)} B_j V_{s(j)}^T \right) V_{s(j)} D(j,p) V_{s(p)}^T \right] \left( R_{s(e)} B_{\bar{s}(p,e)} V_{s(p)}^T \right)^T$$

We can then simplify the expression by multiplying matching eigenvectors and noting that $V_{s(p)} = V_{s(j)}$ since all pathways share the same input correlation:

$$\tau \frac{d}{dt} B_e = B_{\bar{t}(p,e)} \left[ S(p) - \left( B_p + \frac{1}{2} \sum_{j \neq p \in \mathcal{T}(t(p))} U_{t(p)}^T U_{t(j)} B_j \right) D(p) \right] B_{\bar{s}(p,e)}$$

This is usually where we would need to stop for the GDLN reduction and Section 4 where the output matrix has different structure, the "race" dynamics are obtain using this reduction. However, in Section 5 because the structure is symmetric we can exploit some more information in this task to

continue the derivation further to a closed form solution. Having determined the effective datasets for each pathway we know that $U_{t(p)}^T U_{t(j)} = -\frac{1}{2}I \; \forall \; j \neq p$. Thus:

$$\tau\frac{d}{dt}B_e = B_{\bar{t}(p,e)}\left[S(p) - \left(B_p + \frac{1}{2}\sum_{j\neq p\in\mathcal{T}(t(p))} -\frac{1}{2}B_j\right)D(p)\right]B_{\bar{s}(p,e)}$$

Finally, we note that due to the symmetry of the task between contexts, all pathways will learn identical singular values. Thus $B_p = B_j \; \forall \; j \neq p \in \mathcal{T}(t(p))$. Thus, we substitute this equality into the dynamics reduction:

$$\tau\frac{d}{dt}B_e = B_{\bar{t}(p,e)}\left[S(p) - \left(B_p - \frac{1}{4}\sum_{j\neq p\in\mathcal{T}(t(p))} B_p\right)D(p)\right]B_{\bar{s}(p,e)}$$

$$= B_{\bar{t}(p,e)}\left[S(p) - \frac{1}{2}B_pD(p)\right]B_{\bar{s}(p,e)}$$

We note that for each pathway our network has two layers: $e \in \{0, 1\}$. Thus:

$$\tau\frac{d}{dt}B_0 = B_{\bar{t}(p,0)}\left[S(p) - \frac{1}{2}B_pD(p)\right]B_{\bar{s}(p,0)}$$

$$= B_1\left[S(p) - \frac{1}{2}B_pD(p)\right]I$$

and

$$\tau\frac{d}{dt}B_1 = B_{\bar{t}(p,1)}\left[S(p) - \frac{1}{2}B_pD(p)\right]B_{\bar{s}(p,1)}$$

$$= I\left[S(p) - \frac{1}{2}B_pD(p)\right]B_0$$

Assuming balanced solutions, which is reasonable from small initial weights we know that $B_0 = B_1$. We may also then switch to consider the dynamics of an entire pathway and not just one layer in the pathway: $B_p = B_1B_0$. The dynamics of the pathway can be obtain by the product rule:

$$\tau\frac{d}{dt}B_p = B_0(\tau\frac{d}{dt}B_1) + B_1(\tau\frac{d}{dt}B_0)$$

$$= B_0B_0\left[S(p) - \frac{1}{2}B_pD(p)\right] + B_1B_1\left[S(p) - \frac{1}{2}B_pD(p)\right]$$

$$= B_p\left[S(p) - \frac{1}{2}B_pD(p)\right] + B_p\left[S(p) - \frac{1}{2}B_pD(p)\right]$$

$$\tau\frac{d}{dt}B_p = 2B_p\left[S(p) - \frac{1}{2}B_pD(p)\right]$$

This is a separable differential equation which can be solved as per the linear dynamics (Saxe et al., 2014; 2019). Thus the full learning trajectory for the $\alpha$-th mode of a a context dependent pathway is (we have removed the dependence on p to lighten notation):

$$B_\alpha(t) = \frac{(2S_\alpha/D_\alpha)}{1 - (1 - \frac{S}{DB^0})*\exp(2S_\alpha\frac{t}{\tau})}$$

This is the equation used to obtain the "closed" dynamics in Section 5 for the three context case.

# G  GDLN DYNAMICS FOR FOUR AND FIVE CONTEXTS WITH HIERARCHICAL STRUCTURE

## G.1  GDLN DYNAMICS FOR FOUR CONTEXTS

We now derive the dynamics when there are four context specific portions of the output space. This closely follows the derivation of the three context case from Appendix F, however we present it here

in full. Specifically, we have a dataset $D = (\mathcal{I}, C, Y)$ with object, context and output feature matrices $\mathcal{I} \in \mathbb{R}^{N \times N}$, $C \in \mathbb{R}^{4 \times N}$ and $Y \in \mathbb{R}^{2N-1 \times N}$ respectively. Here $N$ denotes the number of datapoints. We concatenate the object and context features vertically to form the input matrix: $X \in \mathbb{R}^{N+4 \times N}$. Every object (depicted by a one-hot input vector in $\mathcal{I}$) can be encountered in every context (depicted by another one-hot vector in $C$). It is convenient to partition the input matrix into three blocks with the same one-hot context vector: $X = [X_1, X_2, X_3, X_4]$ where $X_i \in \mathbb{R}^{N+4 \times N/4}$. Similar partitions follow for $\mathcal{I}$ and $C$ separately. Both models are told which of the objects to produce the features for as well as in which context the object is encountered in. The models must then produce the correct output features, some of which are appropriate for all contexts and others only in one specific context.

Our ReLN trained to imitate the ReLU network loss trajectory and dynamics is composed of four pathways. All pathways have a linear layer of 100 neurons resulting in the same number of total hidden neurons as the ReLU network. The first pathway receives all objects and contexts and maps to the full output matrix. We call this the common pathway and denote it by $*$, with weight and output mapping: $W_* = \{W_*^1, W_*^2\}$ and $\hat{y}^* = W_*^2 W_*^1 X$. Thus, to model this pathway's loss trajectory we apply the linear dynamics equations of Saxe et al. (2014) to the input-output correlation: $\Sigma^* = YX^T = U_* S_* V_*^T$. The common pathway's dynamics then follow $W_*^2 W_*^1 = U_* A_*(t) V_*^T$ and will have the mapping $\hat{Y}^*(t) = U_* A(t)_* V_*^T X$ at all points in time. It is assumed that the network aligns to the dataset singular vectors immediately with $U_*$ and $V_*^T$. The dynamics of the network can then be obtained through the decoupled singular value dynamics of:

$$\pi_\alpha(t) = \frac{S_\alpha/D_\alpha}{1 - (1 - \frac{S_\alpha}{D_\alpha \pi_0}) \exp(\frac{-2S_\alpha}{\tau} t)}$$

where $S_\alpha$ is the $\alpha$-th mode of the input-output correlation singular value matrix $S$, $D_\alpha$ is similarly the $\alpha - th$ mode of the input correlation matrix $(XX^T)$ and $\pi_0$ is the initial state of the network singular values $\pi_\alpha$. Thus, $\pi_\alpha$ begins with a value of $\pi_0$ and increases until its convergence point at $S_\alpha/D_\alpha$. We can then define the residual of this model as $Y^* = Y - \hat{Y}^*(\infty)$.

The three remaining (context specific) pathways are then trained on this residual. The context specific pathways are each gated on for two context settings and connect to all output features. In addition, while these pathways are used in two contexts they are not shown the input context as a feature. The result is that these pathways can only use the input features $\mathcal{I}_i$ in their mapping when they are gated on. Since these pathways are trained at once and have dynamics which depend on the other pathways we now use the GDLN framework (Saxe et al., 2022) to determine their dynamics (we review this setup in Section 2 in the main text). The learning rule for a weight matrix in a GDLN follows the update equation (we use the brackets $\langle \cdot \rangle$ to denote averaging as typically used in statistical physics to simplify notation):

$$\tau \frac{d}{dt} W_e = -\frac{\delta L(\{W\})}{\delta W_e} \forall e \in E$$
$$= \left\langle \sum_{p \in P(e)} g_p W_{\bar{t}(p,e)}^T \left[ y_{t(p)} x_{s(p)}^T - h_{t(p)} x_{s(p)}^T \right] W_{\bar{s}(p,e)}^T \right\rangle_{y,x,g}$$

We note that for our architecture there is only one pathway that each edge is involved in. Thus we may drop the summation over pathways and denote the one relevant pathway with $p$. We also apply the usual linearity of the expectation:

$$= W_{\bar{t}(p,e)}^T \left[ \left\langle g_p y_{t(p)} x_{s(p)}^T \right\rangle_{y,x,g} - \left\langle g_p h_{t(p)} x_{s(p)}^T \right\rangle_{y,x,g} \right] W_{\bar{s}(p,e)}^T$$

The next step is to determine the output of the network which impacts pathway $p$. Specifically we substitute in for $h_{t(p)}$ the sum of all pathways mapping onto $t(p)$ - the terminal point for pathway $p$:

$$h_{t(p)} = \sum_{j \in \mathcal{T}(t(p))} g_j W_j x_{s(j)}$$

Here $\mathcal{T}(t(p))$ is the set of all pathways leading to the same terminal node as pathway $p$. Thus $j$ denotes each of these pathways with corresponding mapping $W_j$ (note $W_j$ here is the product of all

matrices along pathway $j$). Finally $x_{(s(j))}$ is the input to pathway $j$ since $s(j)$ denotes the source of $j$. Substituting this into the dynamics equation we obtain:

$$= W_{\bar{t}(p,e)}^T \left[ \left\langle g_p y_{t(p)} x_{s(p)}^T \right\rangle_{y,x,g} - \left\langle g_p \sum_{j \in \mathcal{T}(t(p))} g_j W_j x_{s(j)} x_{s(p)}^T \right\rangle_{y,x,g} \right] W_{\bar{s}(p,e)}^T$$

$$= W_{\bar{t}(p,e)}^T \left[ \left\langle g_p y_{t(p)} x_{s(p)}^T \right\rangle_{y,x,g} - \sum_{j \in \mathcal{T}(t(p))} W_j \left\langle g_j x_{s(j)} x_{s(p)}^T g_p \right\rangle_{y,x,g} \right] W_{\bar{s}(p,e)}^T$$

$$= W_{\bar{t}(p,e)}^T \left[ \Sigma^{yx}(p) - \sum_{j \in \mathcal{T}(t(p))} W_j \Sigma^x(j,p) \right] W_{\bar{s}(p,e)}^T$$

Thus, the dynamics for one layer of weights ($W_e$) in a single pathway ($p$) depends on the input-output correlation for that pathway ($\Sigma^{yx}(p)$), the input correlation for the pathway itself ($\Sigma^x(p,p)$) and the input correlation for the pathway with the input to all other pathways sharing the terminal point ($\Sigma^x(j,p); j \neq p$). Thus we rewrite the dynamics in terms of these three kinds of correlations separately:

$$= W_{\bar{t}(p,e)}^T \left[ \Sigma^{yx}(p) - W_p \Sigma^x(p,p) - \sum_{j \neq p \in \mathcal{T}(t(p))} W_j \Sigma^x(j,p) \right] W_{\bar{s}(p,e)}^T$$

We note then that for the particular setting we are working in $\Sigma^x(j,p) = \frac{2}{3}\Sigma^x(p,p)$. Thus, we denote $\Sigma^x = \Sigma^x(p,p)$ and substitute this relationship into the dynamics above. Now the input correlation terms do not depend on the alternative pathways taken to the terminal points.

$$= W_{\bar{t}(p,e)}^T \left[ \Sigma^{yx}(p) - W_p \Sigma^x - \sum_{j \neq p \in \mathcal{T}(t(p))} W_j \frac{2}{3}\Sigma^x \right] W_{\bar{s}(p,e)}^T$$

$$= W_{\bar{t}(p,e)}^T \left[ \Sigma^{yx}(p) - W_p \Sigma^x - \left( \frac{2}{3} \sum_{j \neq p \in \mathcal{T}(t(p))} W_j \right) \Sigma^x \right] W_{\bar{s}(p,e)}^T$$

$$\tau \frac{d}{dt} W_e = W_{\bar{t}(p,e)}^T \left[ \Sigma^{yx}(p) - \left( W_p + \frac{2}{3} \sum_{j \neq p \in \mathcal{T}(t(p))} W_j \right) \Sigma^x \right] W_{\bar{s}(p,e)}^T$$

We now apply the change of variables to obtain the dynamics reduction:

$$\tau \frac{d}{dt} \left( R_{t(e)} B_e R_{s(e)^T} \right) = \left( U_{t(p)} B_{\bar{t}(p,e)} R_{t(e)}^T \right)^T \left[ U_{t(p)} S(p) V_{s(p)}^T - \right.$$

$$\left. \left( U_{t(p)} B_p V_{s(p)}^T + \frac{2}{3} \sum_{j \neq p \in \mathcal{T}(t(p))} U_{t(j)} B_j V_{s(j)}^T \right) V_{s(j)} D(j,p) V_{s(p)}^T \right] \left( R_{s(e)} B_{\bar{s}(p,e)} V_{s(p)}^T \right)^T$$

We can then simplify the expression by multiplying matching eigenvectors and noting that $V_{s(p)} = V_{s(j)}$ since all pathways share the same input correlation:

$$\tau \frac{d}{dt} B_e = B_{\bar{t}(p,e)} \left[ S(p) - \left( B_p + \frac{2}{3} \sum_{j \neq p \in \mathcal{T}(t(p))} U_{t(p)}^T U_{t(j)} B_j \right) D(p) \right] B_{\bar{s}(p,e)}$$

Once again because the dataset structure is symmetric we can exploit some more information in this task to continue the derivation further to a closed form solution. Having determined the effective datasets for each pathway we know that $U_{t(p)}^T U_{t(j)} = -\frac{1}{6}I \; \forall \; j \neq p$. Thus:

$$\tau \frac{d}{dt} B_e = B_{\bar{t}(p,e)} \left[ S(p) - \left( B_p + \frac{2}{3} \sum_{j \neq p \in \mathcal{T}(t(p))} -\frac{1}{6} B_j \right) D(p) \right] B_{\bar{s}(p,e)}$$

Finally, we note that due to the symmetry of the task between contexts, all pathways will learn identical singular values. Thus $B_p = B_j \; \forall \, j \neq p \in \mathcal{T}(t(p))$. Thus, we substitute this equality into the dynamics reduction:

$$\tau \frac{d}{dt} B_e = B_{\bar{t}(p,e)} \left[ S(p) - \left( B_p - \frac{1}{9} \sum_{j \neq p \in \mathcal{T}(t(p))} B_p \right) D(p) \right] B_{\bar{s}(p,e)}$$

$$= B_{\bar{t}(p,e)} \left[ S(p) - \frac{1}{3} B_p D(p) \right] B_{\bar{s}(p,e)}$$

We note that for each pathway our network has two layers: $e \in \{0, 1\}$. Thus:

$$\tau \frac{d}{dt} B_0 = B_{\bar{t}(p,0)} \left[ S(p) - \frac{1}{3} B_p D(p) \right] B_{\bar{s}(p,0)}$$

$$= B_1 \left[ S(p) - \frac{1}{3} B_p D(p) \right] I$$

and

$$\tau \frac{d}{dt} B_1 = B_{\bar{t}(p,1)} \left[ S(p) - \frac{1}{3} B_p D(p) \right] B_{\bar{s}(p,1)}$$

$$= I \left[ S(p) - \frac{1}{3} B_p D(p) \right] B_0$$

Assuming balanced solutions, which is reasonable from small initial weights we know that $B_0 = B_1$. We may also then switch to consider the dynamics of an entire pathway and not just one layer in the pathway: $B_p = B_1 B_0$. The dynamics of the pathway can be obtain by the product rule:

$$\tau \frac{d}{dt} B_p = B_0 (\tau \frac{d}{dt} B_1) + B_1 (\tau \frac{d}{dt} B_0)$$

$$= B_0 B_0 \left[ S(p) - \frac{1}{3} B_p D(p) \right] + B_1 B_1 \left[ S(p) - \frac{1}{3} B_p D(p) \right]$$

$$= B_p \left[ S(p) - \frac{1}{3} B_p D(p) \right] + B_p \left[ S(p) - \frac{1}{3} B_p D(p) \right]$$

$$\tau \frac{d}{dt} B_p = 2 B_p \left[ S(p) - \frac{1}{3} B_p D(p) \right]$$

This is a separable differential equation which can be solved as per the linear dynamics (Saxe et al., 2014; 2019). Thus the full learning trajectory for the $\alpha$-th mode of a a context dependent pathway is (we have removed the dependence on p to lighten notation):

$$B_\alpha(t) = \frac{(3 S_\alpha / D_\alpha)}{1 - (1 - \frac{S}{DB^0}) * \exp(2 S_\alpha \frac{t}{\tau})}$$

This is the equation used to obtain the "closed" dynamics in Section 5 for the four context case.

## G.2 GDLN DYNAMICS FOR FIVE CONTEXTS

An almost identical derivation for the three and four context cases holds for the five context case. We omit the initial portion for brevity as we once again train a linear pathway and the context specific pathways on the residual. In the five context case there are now five residual pathways each active for four contexts. We will pick up the derivation where the specifics of the dataset are used to obtain closed form equations from the neural race reduction.

We rewrite the neural race reduction dynamics in terms of these three kinds of correlations separately:

$$= W_{\bar{t}(p,e)}^T \left[ \Sigma^{yx}(p) - W_p \Sigma^x(p,p) - \sum_{j \neq p \in \mathcal{T}(t(p))} W_j \Sigma^x(j,p) \right] W_{\bar{s}(p,e)}^T$$

We note then that for the particular setting we are working in $\Sigma^x(j,p) = \frac{3}{4}\Sigma^x(p,p)$. Thus, we denote $\Sigma^x = \Sigma^x(p,p)$ and substitute this relationship into the dynamics above. Now the input correlation terms do not depend on the alternative pathways taken to the terminal points.

$$= W_{\bar{t}(p,e)}^T \left[ \Sigma^{yx}(p) - W_p\Sigma^x - \sum_{j \neq p \in \mathcal{T}(t(p))} W_j \frac{3}{4}\Sigma^x \right] W_{\bar{s}(p,e)}^T$$

$$= W_{\bar{t}(p,e)}^T \left[ \Sigma^{yx}(p) - W_p\Sigma^x - \left( \frac{3}{4} \sum_{j \neq p \in \mathcal{T}(t(p))} W_j \right) \Sigma^x \right] W_{\bar{s}(p,e)}^T$$

$$\tau\frac{d}{dt}W_e = W_{\bar{t}(p,e)}^T \left[ \Sigma^{yx}(p) - \left( W_p + \frac{3}{4} \sum_{j \neq p \in \mathcal{T}(t(p))} W_j \right) \Sigma^x \right] W_{\bar{s}(p,e)}^T$$

We now apply the change of variables to obtain the dynamics reduction:

$$\tau\frac{d}{dt}\left( R_{t(e)}B_e R_{s(e)^T} \right) = \left( U_{t(p)}B_{\bar{t}(p,e)}R_{t(e)}^T \right)^T \left[ U_{t(p)}S(p)V_{s(p)}^T - \right.$$
$$\left. \left( U_{t(p)}B_p V_{s(p)}^T + \frac{3}{4} \sum_{j \neq p \in \mathcal{T}(t(p))} U_{t(j)}B_j V_{s(j)}^T \right) V_{s(j)}D(j,p)V_{s(p)}^T \right] \left( R_{s(e)}B_{\bar{s}(p,e)}V_{s(p)}^T \right)^T$$

We can then simplify the expression by multiplying matching eigenvectors and noting that $V_{s(p)} = V_{s(j)}$ since all pathways share the same input correlation:

$$\tau\frac{d}{dt}B_e = B_{\bar{t}(p,e)} \left[ S(p) - \left( B_p + \frac{3}{4} \sum_{j \neq p \in \mathcal{T}(t(p))} U_{t(p)}^T U_{t(j)}B_j \right) D(p) \right] B_{\bar{s}(p,e)}$$

Once again because the dataset structure is symmetric we can exploit some more information in this task to continue the derivation further to a closed form solution. Having determined the effective datasets for each pathway we know that $U_{t(p)}^T U_{t(j)} = -\frac{1}{12}I \; \forall \; j \neq p$. Thus:

$$\tau\frac{d}{dt}B_e = B_{\bar{t}(p,e)} \left[ S(p) - \left( B_p + \frac{3}{4} \sum_{j \neq p \in \mathcal{T}(t(p))} -\frac{1}{12}B_j \right) D(p) \right] B_{\bar{s}(p,e)}$$

Finally, due to the symmetry of the task between contexts, all pathways will learn identical singular values. Thus $B_p = B_j \; \forall \; j \neq p \in \mathcal{T}(t(p))$. Thus, we substitute this equality into the dynamics reduction:

$$\tau\frac{d}{dt}B_e = B_{\bar{t}(p,e)} \left[ S(p) - \left( B_p - \frac{1}{16} \sum_{j \neq p \in \mathcal{T}(t(p))} B_p \right) D(p) \right] B_{\bar{s}(p,e)}$$

$$= B_{\bar{t}(p,e)} \left[ S(p) - \frac{1}{4}B_p D(p) \right] B_{\bar{s}(p,e)}$$

We note that for each pathway our network has two layers: $e \in \{0,1\}$. Thus:

$$\tau\frac{d}{dt}B_0 = B_{\bar{t}(p,0)} \left[ S(p) - \frac{1}{4}B_p D(p) \right] B_{\bar{s}(p,0)}$$

$$= B_1 \left[ S(p) - \frac{1}{4}B_p D(p) \right] I$$

and

$$\tau\frac{d}{dt}B_1 = B_{\bar{t}(p,1)} \left[ S(p) - \frac{1}{4}B_p D(p) \right] B_{\bar{s}(p,1)}$$

$$= I \left[ S(p) - \frac{1}{4}B_p D(p) \right] B_0$$

Assuming balanced solutions, which is reasonable from small initial weights we know that $B_0 = B_1$. We may also then switch to consider the dynamics of an entire pathway and not just one layer in the pathway: $B_p = B_1 B_0$. The dynamics of the pathway can be obtain by the product rule:

$$\tau \frac{d}{dt} B_p = B_0 (\tau \frac{d}{dt} B_1) + B_1 (\tau \frac{d}{dt} B_0)$$

$$= B_0 B_0 \left[ S(p) - \frac{1}{4} B_p D(p) \right] + B_1 B_1 \left[ S(p) - \frac{1}{4} B_p D(p) \right]$$

$$= B_p \left[ S(p) - \frac{1}{4} B_p D(p) \right] + B_p \left[ S(p) - \frac{1}{4} B_p D(p) \right]$$

$$\tau \frac{d}{dt} B_p = 2 B_p \left[ S(p) - \frac{1}{4} B_p D(p) \right]$$

This is a separable differential equation which can be solved as per the linear dynamics (Saxe et al., 2014; 2019). Thus the full learning trajectory for the $\alpha$-th mode of a a context dependent pathway is (we have removed the dependence on p to lighten notation):

$$B_\alpha(t) = \frac{(4 S_\alpha / D_\alpha)}{1 - (1 - \frac{S}{DB^0}) * \exp(2 S_\alpha \frac{t}{\tau})}$$

This is the equation used to obtain the "closed" dynamics in Section 5 for the five context case.

## H  HYPER-PARAMETERS FOR CONTEXTUAL DATASETS

Here we provide the hyper-parameters used in Sections 4, 5 and 6. We provide the number of modules needed for each type of GDLN referenced in the main text. Importantly, these modules would still need to be connected to the appropriate portions of the input and output space, as described there. As described in Appendix A, we only require as many hidden neurons as the rank of the input-output correlation matrix the linear pathway is aiming to solve. Finally, as long as the ReLU network and GDLN begin with the same hyper-parameters the mapping to a ReLN will be valid (assuming the correct allocation of linear pathways).

| Hyper-parameter | Value |
|---|---|
| Num-Hidden (GDLN) | 100 |
| Num-Modules (GDLN Single Modules, 3 Contexts) | 4 |
| Num-Modules (GDLN Double Modules, 3 Contexts) | 4 |
| Num-Modules (GDLN Single and Double Modules, 3 Contexts) | 7 |
| Num-Modules (GDLN Triple Modules, 4 Contexts) | 5 |
| Num-Modules (GDLN Quad Modules, 5 Contexts) | 6 |
| Num-Hidden (ReLU) | 700 |
| Num-Modules (ReLU) | NA |
| Init-Scale | $1 \times 10^{-7}$ |
| Num-Epochs | 8000 |
| Step-Size | 0.001 |

Table 1: Hyper-parameters used for the contextual tasks in Sections 4, 5 and 6.

