# OpenReview forum: "Make Haste Slowly: A Theory of Emergent Structured Mixed Selectivity in Feature Learning ReLU Networks"
_ICLR.cc/2025/Conference — ICLR 2025 Poster_

### Official Review · Reviewer_H7x8 · 2024-10-25

**Soundness:** 3
**Presentation:** 3
**Contribution:** 2
**Rating:** 8
**Confidence:** 3

**Summary:**

This paper shows that, under strong alignment assumptions of the data and network weights, finite feature learning ReLU networks learn modules that can be represented by Gated Deep Linear Networks (GDLNs), for which the exact learning dynamics are analytically tractable. For a synthetic task with hierarchical structure, the training dynamics of 2-layer ReLU nets are shown to exactly match those of a GDLN constructed for the task. Through this equivalence, it is shown that ReLU networks learn mixed-selective representations due to a bias for optimal learning speed.

**Strengths:**

The approach of identifying the linear submodules learned by ReLU networks and using this for tractability of the analysis is interesting. At least on toy datasets, natural modules to learn the structure in the data are exactly recovered by ReLU networks together with the learning speed. This provides mechanistic understanding how the bias toward learning speed explains why 2-layer ReLU networks learn common and context-specific but entangled representations.

**Weaknesses:**

I cannot strongly recommend acceptance as some key questions remain unaddressed. While this is a nice and tractable toy model for the learning dynamics of 2-layer ReLU networks, generalizability of the findings is questionable. The results only hold for the synthetic and very specific structure considered. The relevance on real data sets or practical architectures remains elusive, as it is not evaluated. The authors show and acknowledge that the silent alignment Assumption 2 already does not hold for 2-hidden-layer ReLU nets. In Figure 6, against the authors’ claim of ‘sharing the same characteristics’, ReLU networks appear to make larger, sharper steps in the loss. What happens in more practical architectures and real datasets, and whether an appropriate and interpretable GDLN can still be identified remains completely unclear. Experiments that show that this approach works on real data would greatly alleviate these concerns. See other critical questions below.

**Typos in lines:** 30, 392, many in 462-463, 473, 476, 477, 503, 527

**Questions:**

**Critical questions (order: most important to least important):**

1. Your analysis looks very particular to the structure of your synthetic task. Do you think it will be feasible to find unique or representative GDLNs for more complex datasets and architectures?

    a. Are the assumptions more generally broken, even for 2-layer ReLU nets beyond this toy task of hierarchical structure without noise? Atanasov et al. (2021) show that non-whitened data can weaken the silent alignment effect.

    b. ‘Once the common pathway has finished learning there is no correlation between the context features and labels’. But in practice, likely learning is not perfectly separated. Which impact does this have on the generalizability of your results?

2. You claim that the learning dynamics of the ReLN exactly describe those of the ReLU network. Why are the curves in Figure 4 then not exactly matching?
3. How do your results depend on the number of neurons? You never explicitly mention this.
4. Your proposed clustering algorithm in Appendix B evaluates the model at different stages of training. Is this not dangerous in cases where the learned function evolves in a more complex way than in your toy task and systematically changes? Then the algorithm tries to cluster outputs from intermediate functions that are rather unrelated to the final learned function. I would like to see whether clear gating mechanisms can be identified when applied to real data.

**Questions out of curiosity:**

1. In Figures 4 and 5, why is not the largest SV learned first?
2. Where does the bias towards node reuse come from? Can you see this in the learning dynamics equations?
3. Can your analysis be extended to SGD? Would you expect fundamentally different dynamics?
4. Can this theory also inform how to maybe slow down learning to learn disentangled representations?

**References:**

Atanasov, Alexander, Blake Bordelon, and Cengiz Pehlevan. "Neural networks as kernel learners: The silent alignment effect." *arXiv:2111.00034* (2021).

---

> ### Author Response · Authors · 2024-11-18
> **Response to Reviewer H7x8 by Authors (Part 1 of 2)**
>
> We thank the reviewer for their time and consideration of our work. We will begin by addressing the weaknesses in order:
> 1. We kindly direct the reviewer towards the general comment where we discuss the assumption of our paradigms. We thank the reviewer for the question as we recognise that more justification of our assumptions must be added to the text and will strengthen the applicability of our work.
> 2. We thank the reviewer for pointing out the typos and will fix these on a subsequent draft.
>
> Turning to the critical questions:
> 1. We kindly direct the reviewer to the general comment. However, to be more specific to the questions: we show clearly for our setting that the addition of depth breaks the assumption and can conclude that this is due to Lemma 4.1 no longer holding. To our knowledge this clear demonstration that multiple layers of ReLU neurons disrupts feature learning is new to our work. Thus, it is possible that other factors could impact the alignment and we provide a framework for similarly exploring these factors. In Appendix B we discuss our views on how future work could make finding the network gates more feasible for complex datasets and architectures. Similarly it is likely that a dataset could be constructed where the common pathway is not learned first. Once again, the framework we  provide here could be of use in identifying such cases and providing clear insight into why the behaviour changes. A similar phase shift from linearity to nonlinearity is shown for the XoR task in Section 3. Thus, it appears reasonable that future applications of our paradigm could lend similar insight. Importantly, the only general claims of this work is that for every ReLU network there exists a ReLN - which we prove, and that the ReLN framework could be useful to future theorists and shed light on neural network behaviour more broadly. Indeed, one of the largest strengths of the linear neural network paradigm is that it demonstrates how the dataset structure affects learning.
> 2. The ReLU network and ReLNs are initialised with different weights (they are still small as per the linear network framework). This can lead to very minor discrepancies especially as the ReLU network is learning to identify relevant nonlinear pathways which the ReLN is provided from the beginning of training. A similar effect where minor deviations in the randomly initialised values of a linear neural network can lead to minor variation from the closed form dynamics can be seen in Saxe et al. (2019). Finally, on Lines 263 to 267 we do not say we see exact correspondence. Indeed we say “near exact” and “excellent agreement”. In the caption of Figure 4 we say that we see exact agreement between the output of the ReLU network and ReLN at three points in time. We do not mean to be pedantic in our wording here but take the accuracy of our claims seriously and believe these to be accurate descriptions of the results depicted in Figure 4. Similarly, for the concern on Figure 6 saying that the ReLN “shares the same characteristics” as the ReLU network we believe there may be a slight misunderstanding here as the full sentence says “shares the same characteristics in variance”. In other words the points along the trajectories where the drops in loss become inconsistent are the same and have a similar degree of variance. We agree that the ReLU network drops in loss are more stage-like. This point is reiterated on Lines 472 where we say “we also see the same variance profiles”. Indeed, our main claim in Section 6 is one Line 468: “we see that the GDLN still informative about the behaviour of the ReLU network”. We will make clearer our discussion around the variance of the trajectories and commonalities between the dynamics in Section 6. We apologise for the misunderstanding we may have caused and thank the reviewer for pointing this out.
> 3. We thank the reviewer for raising this potential point of confusion and will fix this in a subsequent draft. We direct the reviewer to Lines 252 and 253 of Section 4 in the main text. We say “We use a linear output layer, assuming the model is over-parametrised (a pathway needs to have at least $h$ hidden nodes to learn a rank $h$ effective dataset)”. We will be clear that this is the only consideration for the number of hidden neurons. In addition we direct the reviewer to our revision of the deep linear network paradigm in Appendix A, specifically Lines 808 to 813 where we say: “We also assume that the network has at least $|\Sigma^{yx}|$ hidden neurons (the rank of $\Sigma^{yx}$ which determines the number of singular values in the input-output correlation matrix) so that it can learn the desired mapping perfectly. If this is not the case then the model will learn the top $n_h$ singular values of the input-output mapping where $n_h$ is the number of hidden neurons (Saxe et al., 2014).”

---

> > ### Author Response · Authors · 2024-11-19
> > **Response to Reviewer H7x8 by Authors (Part 2 of 2)**
> >
> > Continuing first with the final critical question:
> >
> > 4\. The reviewer’s point is correct, however Appendix B provides a simple algorithm for obtaining GDLN gates, but also points to meta-learning as a future direction for addressing this limitation (the general comment elaborates more on the use of meta-learning). In addition, we point the reviewer towards Lines 166 and 167 in the main text where we say “Both the gates and weights can be time dependent, however in this section and Sections 4 and 5 the gating structure is constant from the beginning”. Thus, it was valid to assume in the clustering algorithm that gates are consistent over time. Importantly, we do not propose that the clustering algorithm be used for more complex tasks and this is also why we mainly present this algorithm in the appendix. This is in spite of the fact that this is the *first algorithm* which has been proposed for learning the gates of a GDLN (Saxe et al. (2022) assumed the gates were provided with the dataset). Thus, the algorithm could be seen as a contribution, but we agree with the reviewer that it has clear limitations, and so we have not taken this stance. The role the clustering algorithm plays in this work is to provide a proof-of-concept, first step towards learning the gates and also demonstrates a conceptual connection towards meta-learning.
> >
> > Finally the questions out of curiosity:
> > 1. The learning speed of the linear network is only dependent on the singular values of the $\Sigma^{yx}$ matrix but the stable point depends on the first singular values of $\Sigma^{yx}\Sigma^{x^{-1}}$. Thus, depending on the singular values of $\Sigma^x$ this can make a faster mode stabilise lower.
> > 2. Yes, the bias to node reuse can be seen in Equation 3 and Equation 6 as the update to the layer of weights $W_e$ and its decomposition $B_e$ result from a summation over pathways. Thus, all things equal, being involved in more pathways will increase the update applied to a particular layer. We also direct the reviewer to Lines 1035 to 1038 where we review the GDLN dynamics. We state: “the learning dynamics depend both on the input-output correlations of the effective path datasets and the number of paths to which an edge contributes. Ultimately, the paths which learn the fastest from both pressures will win the neural race”. We will also make this statement in the Background in a revised draft and thank the reviewer for pointing us to this.
> > 3. On Line 801 in Appendix A we state: “ It is helpful to note that since we are using a small learning rate the full batch gradient descent and stochastic gradient descent dynamics will be the same”. The same point is made in Saxe et al. (2019). However, it is worth noting the context here is in the continuous time limit of sufficiently small step size with linear network. In practice, minor deviations could be introduced when using SGD compared to full batch learning. It is also possible that the nonlinearity could exacerbate the noise from batching. As we do not explore this point in this work we will refrain from speculating beyond noting that within the linear pathways the dynamics should hold.
> > 4. The two pressures promoting learning speed are the dataset statistics (singular value of the input-output covariance) and the number of pathways a layer is used in. Thus, in theory if a mechanism is defined which creates pathways that have more disentangled feature spaces then learning will be slower. Similarly if the mechanism reduces the number of shared weights then learning will be slower and more disentangled. See Jarvis et al. (2022) for the relation between linear network dynamics and module specialisation (related to disentanglement). This point would lead to our points in Section 7 and Appendix B that the gating pattern can be defined based on rules  different to the ReLU network. Meta-learning the  GDLN with different criteria on the gating patterns could achieve more disentangled representations. Empirically, applying regularisers to the ReLN could also shed light on mechanisms which would help the ReLU networks itself. In summary, yes we believe such insight could be obtained in future work.
> >
> > We thank the reviewer once again for their review and assistance in making the utility of our framework clear. We are happy to engage further to address any remaining questions.

---

> > > ### Comment · Reviewer_H7x8 · 2024-11-25
> > >
> > > I thank the authors for their extensive answers; they have clearly put a lot of thought into their work. I believe the promised changes will improve clarity, in particular around the strong but common assumptions in the deep linear network literature.
> > >
> > > The fact that simple experiments on real data such as on CIFAR10 have not been conducted although 3 out of 4 reviewers asked for them suggests that the gating structure cannot yet be identified or the strong assumptions are not satisfied in practice, and that the ReLN mechanism does not yet extend to realistic settings. Thus the generalizability and practical relevance of the results remains elusive. This limitation is acknowledged in Section 7.
> > >
> > > Still, I believe that the GLDN perspective is very creative and this paper provides the first strong connection to ReLU network dynamics. Whether deeper insights into network dynamics can be drawn from the GLDN framework even if the assumptions are not satisfied, can be explored in future work. The theoretical contribution of showing exact learning dynamics on a non-trivial distribution is an important step toward connecting GLDNs to non-linear networks that provides a complementary perspective for understanding their properties. Therefore, I recommend acceptance despite the acknowledged limitations and have updated my score accordingly.

---

> > > > ### Author Response · Authors · 2024-11-26
> > > > **Response by Authors**
> > > >
> > > > We thank the reviewer again for their time and consideration of our work. We are also very grateful for the consideration given to our rebuttal and the positive assessment of our work. We will ensure that we use the reviewer's feedback to improve on a subsequent draft of our paper.

---

### Official Review · Reviewer_zKcn · 2024-11-03

**Soundness:** 2
**Presentation:** 3
**Contribution:** 2
**Rating:** 5
**Confidence:** 3

**Summary:**

The paper introduced "Rectified Linear Network" (ReLNs) as a subset of Gated Deep Linear Networks (GDLN), and showed that for single-hidden layer network, for a ReLU network, it's possible to find a ReLN that have same loss and outputs at all time-steps (provided that the assumption 2.1 at L159 are satisfied) in a simple synthetic dataset. Using the ReLN as the equivalence of ReLU, the authors provided an analytical solution for the training dynamic for the finite single-hidden layer network with synthetic dataset, and also demonstrated that the predicted loss of ReLN matches with the empirical loss of ReLU networks. In this specific synthetic dataset (which includes hierarchical structure (animals, plants, fish, birds), and multiple contexts), the papers show that the equivalent ReLN network employs an implicit bias for structured mixed selectivity (e.g, one gating pathway can encode multiple context).

**Strengths:**

**Originality**
1. The paper presents a novel framework Rectified Linear Network to analytically investigate the dynamic of finite-width single-hidden layer neural networks via an extension of Gated Deep Linear Network.

**Quality**
1. The paper provides theoretical proof for the equivalence between ReLN and ReLU for the case of finite-width single-hidden layer network and verified that the predicted loss matches with the empirical loss in a synthetic dataset.

**Clarity**
1. The paper clearly present the background relevant work, including the formulation of Gated Deep Linear Network, decomposition of eigenmodes, and derived of learning dynamic based on these eigenmodes.
2. The theoretical proofs are presented clearly with both informal and formal versions, along with a proof sketch in the main paper, and a detailed proof in the appendix, facilitating the reader's understanding
3. The empirical results (including the dataset and figures) are clearly presented with clear figure and caption, along with descriptions in the main text.

**Significance**
1. The paper showed that there's an equivalent ReLN for ReLU for a single-hidden layer network, and it's possible to identify the gating structure of this equivalent ReLN via a clustering algorithm.
2. By investigating the gating structure of the equivalent ReLN network, the paper shows that there's an implicit bias for structured mixed selectivity (e.g, one gating pathway can encode multiple context).

**Weaknesses:**

While the paper proposed a novel framework ReLN to study learning dynamic and feature learning in finite-width neural network, a significant topic in the machine learning community, the paper has several limitations in both theoretical and empirical results, which I would clarify below:

1. Theoretical results: The main theoretical results of the paper requires very strong assumptions (L159, *Assumption 2.1*) in both (1) input dataset (*The dataset correlation matrices are mutually diagonalizable*) and (2) model training trajectory (*The neural network weights align to the singular vectors of the dataset correlation matrices.*). These are very strong assumptions, and the authors did not offer any analysis on specific scenario in which these assumption holds or not hold. As we see in section 6, assumption (2) is violated in the case of 2-layer hidden network. As the paper only uses a very simple synthetic dataset (one-hot vector input and sparse binary-vector output), it's difficult to tell whether assumption (1) can hold for realistic dataset with complicated distribution, even in the 1-layer hidden network case.

2. Empirical evaluations: The empirical experiments that supports the claim and theoretical results only includes single-layer hidden network on a simple synthetic dataset (the paper did include 2-layer hidden network but this experiment did not align with the theoretical results, due to violation of the theoretical assumption). While it's reasonable to use this simple synthetic dataset as a proof of concept to verify the theoretical results and illustrate the mixed-selectivity phenomenon, it would be helpful and more convincing if the authors can demonstrate that the proposed approach work in a realistic image dataset (MNIST, CIFAR, etc.) with a more realistic and variety of model architectures (multiple layer perceptron, convolutional neural nets, etc.). The inclusion of real dataset and variety of architecture is even more important since the theoretical results require such strong assumptions.

3. Identification of gating structure: The gating structure identification is a central bottleneck of this framework, since gating *g* is treated as inputs, and needed to be identified before training. Identifying a fixed gating structure, or varying gating structure through training, would be one of the major difficulty of the framework. The paper did propose a clustering algorithm to identify the gating structure of ReLN from the representation of ReLU models. However, since the paper only operates with very small synthetic dataset and models, it's unclear whether this clustering algorithms can scale with realistic dataset and larger models. Therefore, this is another reason that it is critical to evaluate this proposed framework on more realistic dataset, instead of only on the simple synthetic dataset.

**Questions:**

**Questions**
1. [L159] Would be helpful to comments on cases for each of the mentioned assumptions in `Assumption 2.1` since these seem to be not very general assumptions and can be easily violated.
2. [L183] What is the network architecture for the XoR dataset? Is it a 2-layer network? How would the gating structure look like? It would be helpful to explicitly write down the equation or numerical examples instead of vague description of “2 pathways” and “4 pathways”.
3. [L257] How do these pathways get identified? Are they identified manually based on the prior knowledge about the task? Since identifying gates would be a major bottleneck for the applications of ReLN, it would be helpful to provide more information on how to identify the pathways.

**Suggestions**
1. In the Contribution section, clearly state the models and dataset in which the theoretical and empirical results are obtained (single-layer network for theoretical results, and simple synthetic dataset with hierarchy and context (maybe would be helpful to give this dataset a name for easy reference?) to provide a clear expectation for the readers.
2. Spending more space in the main text to explain how to identify the gating structure and pathways from ReLU to ReLN.

---

> ### Author Response · Authors · 2024-11-18
> **Response to Reviewer zKcn by Authors**
>
> We thank the reviewer for their time and consideration of our work. We will begin by kindly directing the reviewer to the general comment as we discuss all of the noted weaknesses there. Considering the questions:
> 1. We thank the reviewer for the question as we recognise that this justification of our assumptions must be added to the  text and will strengthen the applicability of our work. We kindly direct the reviewer towards the general comment where we discuss Assumption 2.1.
> 2. We thank the reviewer for raising this and will aim to make the network architecture more clear. We direct the reviewer to Appendix C for the full derivation of the setting and dynamics of Section 3. We also note that Figure 2b and 2c depict the network architecture of the ReLNs which imitate the ReLU network with one hidden layer. Each square in these diagrams corresponds to a layer of neurons as described in Section 2, specifically Figure 1. We will be clear in the main text that the ReLU network has a single hidden layer. Further, if the reviewer could perhaps guide us on why Figure 2b and 2c did not have the desired effect (as these aim to depict the gating structure and make clear what is meant by 2 pathways and 4 pathways) then we would be happy to incorporate those changes and suggestions.
> 3. In this case it is possible to identify the gates manually. For the larger networks the clustering algorithm can be used to assist with finding the gates (for the tasks we consider in this work). Please see the general comment and Appendix B for more discussion on how to identify gating structures for other architectures and tasks. Importantly, the manner in which the gates are found is not the focus of Section 3. Rather the insight which the ReLN paradigm offers in demonstrating clearly how a ReLU network rapidly changes gating strategy due to subtle change in dataset structure (forming a phase transition), is the primary aim and contribution of that section (as well as introducing the ReLN paradigm in an intuitive setting).
>
> We thank the reviewer for their suggestions and will certainly incorporate these into a subsequent revision. We thank the reviewer once again for their review and assistance in making the utility of our framework clear. In particular we are thankful for the suggestion to elaborate on the nuance of our assumptions as we believe this addition greatly strengthens the broader applicability of our work. We are happy to engage further to address any remaining questions during the discussion period.

---

### Official Review · Reviewer_YprU · 2024-11-03

**Soundness:** 2
**Presentation:** 2
**Contribution:** 2
**Rating:** 3
**Confidence:** 4

**Summary:**

This paper introduces Rectified Linear Networks (ReLNs), a subset of Gated Deep Linear Networks (GDLNs) designed to capture the training dynamics of finite-dimensional ReLU networks. By drawing an equivalence between ReLU and ReLN, the authors aim to provide theoretical insights into feature learning and structured mixed selectivity in ReLU networks, especially in tasks involving multi-contextual inputs.

**Strengths:**

1. The introduction of ReLN as a GDLN variant to study ReLU dynamics is innovative and offers a new angle on understanding feature learning in finite ReLU networks.
2. The paper provides theoretical insights into inductive biases in ReLU networks, particularly regarding structured mixed selectivity and node reuse.

**Weaknesses:**

1. It remains unclear why ReLU’s training dynamics and mixed selectivity properties cannot be derived directly from ReLU networks, as they’re already nonlinear.
2. The approach is demonstrated on relatively simple tasks, raising concerns about scalability. For instance, applying ReLNs to realistic datasets like MNIST remains unaddressed.
3. Terms like "feature learning" and "pathway doing feature learning" (line 302) lack precise definitions. More clarity is needed to distinguish “feature learning” in this theoretical context.

**Questions:**

1. Could the main findings (e.g., mixed selectivity) be directly observed in the ReLU network without using ReLNs?
2. Does the method extend efficiently to larger datasets (e.g., ReLU networks trained on MNIST), and can a ReLN adequately explain such networks?
3. Could the authors clarify their definition of "feature learning" and what they mean by a pathway "doing feature learning" (line 302)?

---

> ### Author Response · Authors · 2024-11-18
> **Response to Reviewer YprU by Authors (Part 1 of 2)**
>
> We thank the reviewer for their time and consideration of our work. We will begin by addressing the weaknesses in order:
> 1. The nonlinearity of the ReLU network is exactly why the dynamics are intractable (for general datasets and in the feature learning regime - see our response to Weakness 3 for the meaning of feature learning in this work). We direct the reviewer to Appendix A, and specifically Line 856. Here we are reviewing the deep linear network paradigm and are using the fact that for a linear network $Y =W^2W^1X$ we are able to compute the matrix multiplication up front $W=W^2W^1$ providing the network mapping $Y=WX$ (we note that the dynamics of $Y=WX$ and $Y =W^2W^1X$ are different which is why we must begin the derivation from $Y =W^2W^1X$ and cannot just start with $Y =WX$ -  we note this fact on Lines 136 to 139). The ability to resolve $W=W^2W^1$ is due to the associativity of matrix multiplication. However, when we introduce a nonlinearity (denoted as $f(\cdot)$ here) then $Y = W^2f(W^1X)$ and to evaluate $W^2f(W^1X)$ we must know the datapoints $X$. Thus, we cannot summarise the entire network mapping in terms of its singular values any longer. This is why we believe the correspondence we introduce in this work between ReLU networks and ReLNs is useful - it provides a nonlinear network where we are able to map the ReLU network onto individual linear pathways where the *linear network dynamics are once again valid*.
> 2. We kindly direct the reviewer towards the general comment for the broader discussion on the scalability and utility of our approach.
> 3. We will certainly be clearer on this in the  background and direct the reviewer to the example in Appendix A reviewing the linear network dynamics. Specifically Lines 905 to 914. To be clear here, the alignment of the network singular vectors to the singular vectors of the dataset correlation matrices corresponds to feature learning in a dense network. The dot product between two vectors provides a measure of the similarity between the vectors. Thus, when a layer of the network aligns its singular vector to a singular vector of the dataset it means that the layer is looking for that feature (higher similarity in the dot product will result in more activation being propagated through the network down the corresponding hidden neurons). This is conceptually identical to a convolution kernel responding to an appropriate feature as it is applied to an image, except the metric of similarity here is the dot product. The terminology of the feature learning regime versus lazy learning was introduced on Lines 42 and 43 and refers to the two identified training regimes for deep neural networks in the literature (Chizat et al., 2019). The lazy regime occurs when the network begins with large weights and the randomly initialised weights are used to learn the task. We operate in the feature learning regime where the randomly initialised network weights change significantly in aligning with the dataset singular vectors. We note that some prior theoretical work such as the NTK only work explicitly in the lazy regime (Jacot et al., 2018; Lee et al., 2019). Thus, the fact that our work applies to the feature learning regime is useful, and in addition justifies Assumption 2.1.2. Near line 143 we will emphasise that this alignment of the network singular vectors constitutes feature learning and thank the reviewer in helping with the clarity of our work.

---

> > ### Author Response · Authors · 2024-11-18
> > **Response to Reviewer YprU by Authors (Part 2 of 2)**
> >
> > We now turn to the questions:
> > 1. Mixed selectivity, and similar very related phenomena such as polysemanticity (see Section 7 and Appendix B for connections to empirical studies) have been identified empirically. Our paradigm provides rigour however, for example in proving that no other strategy than mixed selectivity would minimise the loss as quickly for our task and architecture and that no other strategy would share the same learning dynamics. There is a long history of theory informing experiment, and experiments guiding theory and we see this work as continuing this trend. Theoretical results provide certainty and clarity but typically have limiting assumptions, while experiments can provide similar insights and scale far more quickly, but have less certainty and interpretability. Indeed, a primary benefit of the deep linear network paradigm is its tractability and interpretability. A motivation of this work is that mixed-selectivity, polysemanticity, modularity and disentanglement are concepts identified in the literature empirically but could still benefit from more theoretical treatment. This connection to established but primarily empirical concepts and results is a strength of this work.
> > 2. We kindly direct the reviewer towards the general comment regarding the scalability of our approach, as well as to the comments made under Weakness 2 above.
> > 3. We kindly direct the reviewer to our comments above under Weakness 3.
> >
> > We thank the reviewer again for their review and assistance in making the utility of our framework clear. We are committed to continuing the discussion if needed, and aim to address any further questions the reviewer may have.

---

> > > ### Comment · Reviewer_YprU · 2024-11-26
> > >
> > > Thank you for addressing my questions.
> > >
> > > Regarding scalability, I want to clarify my point about the generality of paradigms and experiments. My concern is not whether this paradigm could yield multiple follow-up papers in toy settings. Rather, I am asking about its applicability to more complex and realistic datasets beyond simple synthetic settings, such as CIFAR-10 (or even just MNIST). For instance, neural tangent kernels (NTKs) not only provide analytical characterizations of a wide range of realistic models (albeit in certain regimes like the lazy regime) but also offer analytical tools for analyzing representations, such as NTK alignment. My question is: What is the authors' vision for the broader applicability of this paradigm beyond enabling a few more papers on (deep) linear networks?
> > >
> > > I appreciate the hope of leveraging analytical characterizations of simple models (e.g., linear networks) to inform phenomena in realistic models. However, this line of work on linear networks (even deep ones, though often with constant depth) has been somewhat puzzling to me. Having read several papers by Saxe et al. and attended his talks, I find it challenging to reconcile: (i) the analytical techniques developed in these studies seem inherently limited in their ability to generalize to more complex, realistic settings, and (ii) the emerging phenomena or phases can often be observed simply by simulating the networks directly. I would greatly value the authors’ insights on their longer-term vision for research on linear networks and how they see it advancing our understanding of more realistic problems.

---

> > > > ### Author Response · Authors · 2024-12-02
> > > > **Response to Reviewer YprU by Authors (Future Directions)**
> > > >
> > > > We thank the reviewer for their response and are happy that the reviewer agrees that our work could lead to multiple follow-up papers. Even if these are in toy settings, we see them as beneficial work and think this potential to build from our paper supports its significance.
> > > >
> > > > We will respond in three parts and hope that this effectively provides the vision the reviewer is asking for. Firstly, we just note that the models we study in this work are nonlinear. We trust the reviewer is aware of this and that this is just phrasing. However, the question remarks that the follow-up work using our paradigm will be on linear networks. To clarify, we introduce a paradigm for studying the dynamics of nonlinear networks with ReLU activations. This also underscores our main contribution: this work is taking a step towards making the deep linear network paradigm much more applicable to realistic architectures. When the deep linear network framework was first introduced, it was likely unclear how exactly it would be extended to study ReLU networks. However, after ten years of progress, we have begun to achieve this. With more time, we believe this direction will still become far more applicable to realistic settings. Thus, we agree with the reviewer that this is the goal, and we are taking steps to achieve what the reviewer seeks with this work.
> > > >
> > > > Secondly, we will now aim to answer the question directly on how we see this line of work progressing. We see three high-level paths forward:
> > > > 1. We see the use of toy settings as a very important part of theoretical work (we do not think the reviewer is arguing this point either), and the fact that this line of work will be able to continue to inspire new experiments and explain identified phenomena is certainly important. We believe this is a sufficient contribution to begin with, exemplified by prior work in this direction being published in top-tier venues.
> > > > 2. The second direction we see is improving the ability to find gatings for more complex settings. The clustering algorithm we introduce here is very simple but still effective even on the settings we consider. It is very likely that far more sophisticated methods, supplied with enough data, will be able to scale to more complex datasets and architectures.
> > > > 3. The third direction would be to explore the empirical validity of GDLNs more. Indeed, GDLNs were introduced as a model where the dynamics were tractable. This is an interesting approach as it is opposite to the second direction above (and the direction typically taken by theory in general). Typically, an architecture is found, validated empirically, and then we aim to provide a theory for the architecture. The approach here instead establishes a model where the dynamics are tractable inherently, and it is left to scale this to more tasks. Given the resemblance to a mixture of experts and modular networks, it is plausible that GDLNs can be scaled to more realistic use cases.
> > > >
> > > > Importantly, these three directions are complementary and will likely inform each other. Our work in this paper serves the first two directions.

---

> > > > > ### Author Response · Authors · 2024-12-02
> > > > > **Response to Reviewer YprU by Authors (ReLNs in the context of NTK)**
> > > > >
> > > > > A comparison to the NTK framework might help us contextualise our main points. As the reviewer points out:
> > > > > 1. The NTK is a deeply insightful framework and has yielded clear successes in studying large, realistic architectures, in spite of it treating the lazy regime. We agree completely.
> > > > > 2. We also agree that studying the NTK alignment in cases where the assumptions of the NTK dynamics are violated is very helpful.
> > > > >
> > > > > In light of these points, we can consider the limitations of the linear dynamics:
> > > > > 1. The reviewer notes that the linear networks might be inherently limited to simpler settings, and we would consider this similar to the NTK being inherently limited to the lazy regime. The NTK framework has shown it is possible for a framework to be useful, even with inherent limitations. What is exciting to note is the complementary nature of the NTK framework and the one we develop here. NTKs can treat realistic settings but only in the lazy regime, which hides some of the complexities of feature learning (a hallmark of the practical success of neural networks). Our framework currently treats more simplistic settings but in the feature learning regime with more complex dynamics. The fact that the limitations and strengths of our framework are complementary to other theoretical approaches found in the literature is a strength of our work and a large motivation for us to pursue this line of research.
> > > > > 2. The reviewer notes that many of the phenomena we find can also be seen empirically. We agree; however, experiments are limited in the certainty and explainability they offer. If one observes mixed selectivity in a ReLU network, they cannot claim that the network will always be mixed selective, nor can they say exactly why this behaviour emerged. Using our theory, we can now say that the ReLU network in the setting we considered will always utilise structured mixed selectivity, which is clearly due to a learning speed benefit. It is necessary to have the dynamics reduction from the GDLNs to see this. Once again, in comparison to the NTK regime, we note that to observe the NTK alignment in general, it is also necessary to simulate the network directly. Similar to the analytical tool offered by the NTK from these simulations, by defining a GDLN that shows similar behaviour to a ReLU network, we also offer an analytical tool with improved tractability and explainability.
> > > > >
> > > > > While we certainly do not claim to know that our framework will have the same influence as the NTK, we believe these parallels support our claim that our framework provides a valuable contribution to the field and can be significant - in spite of its current limitations.
> > > > >
> > > > > We hope this helps address the reviewer’s concerns. As far as we can see, our work has taken a step towards more realistic settings by deriving the dynamics of finite feature learning ReLU networks. Thus, our work has a clear “short-term” significance by providing this first step to achieving the vision of treating realistic settings. In the medium term, we are motivated by the successes and influence of the NTK, even in cases where some of its assumptions are violated. The parallels we see with our framework would indicate that in the medium term we have provided an analytical tool for more realistic settings and another theory to explain or direct experiments. Finally, in the long term, our vision is that the three complementary directions we have noted will ultimately lead to a model which can be used in realistic settings and provides highly tractable and interpretable dynamics.

---

### Official Review · Reviewer_sUq5 · 2024-11-06

**Soundness:** 4
**Presentation:** 3
**Contribution:** 3
**Rating:** 8
**Confidence:** 3

**Summary:**

This work builds provides a step towards theory of feature learning in finite ReLU neural networks by building on an equivalence between ReLU networks and Gated Deep Linear Networks (GDLNs). The authors introduce "Rectified Linear Networks" (ReLNs), which are GDLNs specifically designed to imitate ReLU networks, allowing them to derive exact learning dynamics.
The key contributions are:

The introduction of ReLNs as a theoretical framework to analyze ReLU networks, providing tractable dynamics for finite-width networks during feature learning. A demonstration that ReLU networks exhibit an implicit bias towards structured mixed selectivity - where neural units are active across multiple contexts but in structured, reusable ways.
Evidence that this bias towards mixed selectivity and node reuse is amplified when: 1) more contexts are added to the task
and 2) additional hidden layers are included in the network

The authors support their theoretical findings with analytical solutions and empirical demonstrations on several tasks, including an adapted XOR problem and multi-context classification tasks. They show that while ReLU networks aren't biased towards strictly modular or disentangled representations, they do learn structured features that can be somewhat reused across contexts.
The work takes a step towards understanding how and why structured representations emerge in ReLU networks during learning, bridging a gap in theoretical understanding between linear networks and modern deep learning architectures.

**Strengths:**

Disclaimer: I did not study the proofs of the paper, as I was very late reviewing this paper. I will try to find more time in the coming weeks.

I find the paper clearly writtin and well presented, the authors make an effort presenting the dense results in a clear manner. The authors, to the best of my understanding, extend the theory around GDLNs and allow for a more in-depth study of the training dynamics of ReLU networks. I find the exposition of quite toyish tasks enlightning and make the paper more easy to follow.

**Weaknesses:**

I can not judge fairly the novely of the paper, especially its relation to Saxe et a., 2022. For me, it would have been helpful to highlight the novelty a bit more.

**Questions:**

1) Can you comment a bit more on Assumption 2.1 - I know that the mutally diagonalizable structure is quite restrictive, in particular, can you comment on why the tasks you chose follow these assumption(s). Which tasks can you not study, give these assumptions?

2) Can you give a bit more experimental results, especially when training your networks. Maybe a hyperparameter table in the appendix is nice. Can you give more details how you derived hyperparameters when training networks, you for example mention some in Figure 2. I missed if these hps are analytically derived.

3) I find the discussion wrt to compositionality and modularity, you mention "strictly modular" in the abstract, a bit unclear. Can you clarify, or even define in the work, what you mean with this - and then contrast this to your findings. I guess the same applies for an up-front (maybe even in the intro) definition of what you mean with this. I would find it easier to follow the paper, having these things more clearly explained.

---

> ### Author Response · Authors · 2024-11-18
> **Response to Reviewer sUq5 by Authors**
>
> We thank the reviewer for their consideration of our work and positive review. To address the one weakness, we will certainly make our contributions from Saxe et al. (2022) more clear. We note that we aimed to highlight our main contributions on Lines 70 to 81 and will work to improve these points. To recap briefly and at a high level our contributions are:
> - Noting the connection between a ReLU network and a subset of GDLNs which we introduce as the ReLNs and the proof of existence of a ReLN for any dense ReLU network.
> - The design of the tasks in Sections 3 to 6 and derivation of the ReLN dynamics in these settings.
> - The subsequent insight which follows into the behaviour of ReLU networks, and the source of structured mixed-selectivity in these cases.
>
> To answer the questions:
> 1. For Question 1 we direct the reviewer towards our general comment above. We do emphasise that if the assumption is violated there are means of obtaining a dynamics reduction rather than a closed form solution, which is a strategy used in GDLNs (Saxe et al. 2022). Thus, there are strategies for  mitigating the effect of breaking the assumption and continuing the analysis with some loss in tractability and interpretability. For many cases, however, this will be sufficient. We thank the reviewer for the question as we recognise that this justification of our assumptions must be added to the text and will strengthen the applicability of our work.
> 2. We will certainly add a hyper-parameter table in the appendix. In addition, the hyper-parameters are discussed more in Appendix A. For example, the initialization variance and learning rate need to be small and the number of hidden units must be greater than the rank of the input-output correlation matrix. The small learning rate is needed to take the continuous time-limit when solving the dynamics (Line 795) and small initialization variance ensures that the two weight matrices begin with roughly balanced singular values (Line 856). The number of hidden neurons is discussed on Lines 811 to 814 and this just needs to be large enough to have sufficient rank to learn the dataset. Importantly, as long as the same hyper-parameters are used for both the ReLU network and ReLN, and these conditions of the linear dynamics are maintained, then the dynamics in Sections 3, 4 and 5 will match. Thus, the hyper-parameters are fairly arbitrary. We still mention them for reproducibility.
> 3. We thank the reviewer for raising this point and will clarify it in a subsequent draft. Modularity in this work refers to the architectural property where a certain set of hidden neurons are responsive to the same set of particular stimuli. When multiple modules exist they can be composed. For example, take a module which can identify red objects and another which can recognise squares. Through their composition (by using both modules) we can identify red squares. By strict modularity we refer to modules of this nature where an individual module can also be used in isolation. In other words it is *not coupled* with any other modules. In the case of the context specific pathways through our ReLNs in Sections 4 to 6 none of the pathways can be used in isolation - they all require another contextual pathway to be active at the same time to obtain the appropriate output response. This is in spite of the fact that the pathways form clear and interpretable subnetworks which respond consistently to a given set of contexts. We will clarify that this decoupling is a property of interest to us and the literature (Andreas et al., 2016; Andreas, 2018) and thank the reviewer for helping to improve the clarity of our work.
>
> We thank the reviewer again for their review and assistance in making the utility of our framework clear. We are happy to answer any further questions or concerns the reviewer may have.

---

### Author Response · Authors · 2024-11-18
**General Rebuttal by the Authors**

We thank the reviewers for their time and thoughtful reviews. We are pleased that the noted weaknesses of our approach are ones which we raise explicitly ourselves and have considered, or are inherited from prior. While we believe it is important that we are clear on the limitations and assumptions of our approach, we acknowledge that these can raise questions. We will address the common weaknesses and questions in a thread here, and appreciate the reviewers’ assistance with clarifying our assumptions and the utility of our framework. We will begin by summarising our main points and changes. We then elaborate below on these points if more information or explanation is needed.

---

> ### Author Response · Authors · 2024-11-18
> **General Rebuttal by the Authors (Summary of Points)**
>
> The limitations of Assumption 2.1:
> - Assumption 2.1 is necessary for the full tractability and interpretability offered by the deep linear network dynamics. However if it is violated we can still continue the derivation to a dynamics reduction similar to the GDLN framework, which is insightful and likely to be enough for many cases. The fact that Assumption 2.1 being violated does not stop a useful derivation being obtained was not clearly mentioned and will be added to a revised draft.
>
>
> The limitations of Assumption 2.2:
> - In the context of the deep linear network paradigm, Assumption 2.2 is equivalent to saying that the network is successfully feature learning. In Section 6 we show a case where the vectors do not align fully and the consequences this has. The network on Section 6 is certainly doing feature learning as the vectors are still aligning to a significant degree, but from the linear network dynamics perspective there is an element of lazy learning. Being able to provide dynamics which can accommodate imperfect feature learning of this nature without relying on high dimensional limits is an important direction of future work. However, all theories of feature learning will have a similar assumption. We will add this description to the background as it was omitted. This also aids in clarifying that this alignment is our notion of feature learning.
>
>
> The absence of larger scale empirics:
> - Mixed selectivity is an established phenomena in the machine learning and neuroscience literature, especially with recent interest in similar concepts like polysemanticity  (Olah et al., 2017a; Lecomte et al., 2024). In our case distilling this complex phenomenon into a parsimonious setup where theory is manageable is a contribution. Thus, the tractable setting reproducing empirical phenomena is a strength. We will be clearer in Section 3 that mixed selectivity is an established concept but we provide a mechanism from which mixed selectivity may emerge. Specifically, node reuse speeds up learning and can be used to quickly minimise loss. We will also draw this conclusion more clearly from Equations 3 and 6 (shown by the summation over pathways).
>
>
> The difficulty of findings gates:
> - While we provide an algorithm to find the gates in the tasks considered in this work, we agree that the algorithm we present is simple. Hence we do not claim this to be a large contribution of our work. We include the algorithm to demonstrate our recognition of its importance and provide direction for future work. We do note that this is an advance on the GDLN framework which assumed gates were given as part of the dataset (Saxe et al. 2022). We will move some of the discussion on finding gates from Appendix B into Section 7 and discuss in more detail in the main text how gating structures may be found in our framework.
>
>
> The scalability of our paradigm and experiments:
> - On the paradigm generality: The assumptions and limitations of our work are common in prior, accepted theoretical work on the deep linear dynamics paradigm which have been presented at high-impact venues (Saxe et al. 2014,Saxe et al. 2019,Saxe et al. 2022,Jarvis et al. 2023). More importantly the simple datasets which they study have all led to valuable insight into larger models. The same is true of all other paradigms of theory we note in the Introduction, which have various limitations but equally have been highly insightful. We argue that the same will be true of our paradigm, especially considering that our paradigm satisfies an explicit set of assumptions not possible with any other paradigm.
> - On the experiment generality: The tasks we consider here are motivated by prior work which has proven insightful to the field. The extended XoR task we present is a generalisation of the seminal XoR task which is of broad interest to the machine learning community. More importantly it shows a clear phase transition in the use of nonlinearity by a ReLU network on a highly interpretable domain. This makes it ideal for introducing the paradigm and this behavioural phase transition resulting from a gradual change in dataset structure is an insight which to our knowledge has not been shown before. Similarly the contextual task builds on prior work. The individual output structures are all from Saxe et al. (2019) - which shed light on semantic learning in artificial neural networks and children. The input structure, while simple, was based on the manner context was incorporated in older connectionist models  (Rogers et al., 2004), but also more recently with larger deep networks (Sodhani et al., 2021). Ultimately, the datasets presented in Sections 4,5 and 6 are significantly more complex and realistic than any previous datasets used in the deep linear network paradigm. Similar to adding more information on mixed-selectivity in the literature contextualising our results and methods, we will elaborate more on these points in the main text of Section 3.

---

> ### Author Response · Authors · 2024-11-18
> **General Rebuttal by the Authors (Elaboration on Assumptions Part 1 of 2)**
>
> The point where both assumptions, from Assumption 2.1, are used in the deep linear dynamics framework can be seen in Appendix A from Line 824 to 841. Firstly, we note from the equations that the assumptions are independent of the task and necessary for the deep linear network framework as a whole (Saxe et al. 2014). From these equations it is also clear what will occur when the assumptions fail:
>
> For Assumption 2.1.1 on the mutual diagonalisability of the input-output and input correlation matrices:
> - If this does not hold then $\Sigma^{yx} = US\hat{V}^T$ and $\Sigma^x = VDV^T$. Note that now $\hat{V}$ and $V$ are different matrices. In this case the update equations in terms of the singular vectors will be: $\tau \frac{d}{dt} \bar{W}^1 = \bar{W}^2(S\hat{V}^TV - \bar{W}^2\bar{W}^1D)$ and $\tau \frac{d}{dt} \bar{W}^2 = \bar{W}^1(S\hat{V}^TV - \bar{W}^2\bar{W}^1D)$. Thus, there is now a matrix $\hat{V}^TV $ relating the two correlation matrices. It is important to note that these equations are still highly useful reductions to the gradient descent dynamics and the matrix $\hat{V}^TV$ is potentially interpretable (especially in comparison to interpreting weights of the network). However, from the perspective of the deep linear network framework we can no longer obtain fully closed form equations for the training dynamics of the network. Secondly, the exact features learned by the linear network are no longer as clear. Thus, there is merit toward aiming to maintain the full tractability and interpretability of the linear dynamics. For the GDLN framework specifically, the violation of Assumption 2.1.1 is even less severe. For the GDLN framework the closed form dynamics are not always obtainable (in Section 4 we use the reduction dynamics, while in Section 5 we use closed form dynamics and we derive these cases in Appendices F and G) and in fact we will still be able to derive the neural race reduction to the same point shown in Equation 12, just with the added term of $\hat{V}^TV$. Thus, some loss of interpretability for the linear pathways is the consequence of violating the assumption. To summarise - two of the primary strengths of the deep linear network paradigm is its tractability and interpretability. Assumption 2.1.1 is key for both properties. However, if it is violated we are still able to make very meaningful progress in deriving a dynamics reduction in a similar manner to the original GDLN paper and in many cases this is likely to be sufficient (Saxe et al. 2022). In the revised draft we will be clear in the main text that Assumption 2.1.1 is needed for full tractability and analysability but there are means of making clear progress towards a reduction if it is violated. We will also add a more in-detail discussion to Appendix A where the assumptions are introduced in the linear network dynamics review. We believe these two corrections will greatly add to the clarity, but also ensure that we do not make our framework overly restrictive or prescriptive. We thank the reviewers for their assistance with these points.
>
> For Assumption 2.1.2:
> - A failure of this assumption can lead towards variability in the loss trajectory of the model. We clearly demonstrate such a case in Section 6. Further, if this assumption is violated then a reduction, while possible, is still quite complex as seen on Lines 824 to 842. However, in spite of a violation of this assumption a GDLN is still interesting from two perspectives: Firstly the GDLN will likely shed light on some difficulty involved in full feature learning (the network fully aligning to the dataset correlation matrices). Secondly, the GDLN is still able to be used as a comparative model or baseline for the ReLU network - particularly as  it is more interpretable with the explicit gating pattern. Finally, Assumption 2.1.2 is necessary as we aim to study feature learning. Interesting cases can arise where there is significant movement towards alignment in feature space but not convergence (we believe the demonstration of such an effect in Section 6 and the insight it lends to a ReLU network is a contribution of this work), however, if this assumption is severely broken (the modes align very little or not at all) then this would mean that the analysis is no longer considering feature learning. This defeats the original purpose of our proposed framework - to begin to address the gap in theoretical paradigms for finite size, feature learning, nonlinear (with ReLU activation) models. As we motivate in the Introduction, we believe finite size, feature learning, ReLU network are of practical interest to the field and should be considered theoretically, which would necessitate Assumption 2.1.2 or something of a similar nature. We will clarify the purpose of the assumptions in the main text and add a similar discussion to the one here into our review of the linear dynamics framework. We thank the reviewers again for assisting with the clarity of our assumptions.

---

> > ### Author Response · Authors · 2024-11-18
> > **General Rebuttal by the Authors (Elaboration on Assumptions Part 2 of 2)**
> >
> > Briefly, we can comment on when Assumption 2.1.1 will not hold. A counter-example to the assumption can be derived in a rank-2 setting for an arbitrary $X$ where the inclusion of $Y$ in the input-output correlation calculation reverses the ordering of the right singular vectors $V$ compared to the right singular vectors of the input-correlation matrix. In this case the $\hat{V}^TV$ will remain in the dynamics reduction and be a matrix with $1$'s on the off-diagonals and $0$'s on the diagonals to accommodate the reversal of the modes. To achieve this though the individual values in the Y matrix must be very irregular and arguably unnatural. However, it is also possible to construct far more complex datasets where the assumption holds (we do so in this work). What this implies is that there is no brief statement on when the assumption will apply. It depends entirely on design decisions of encoding the dataset and the task at hand. The point is not whether counter-examples can be constructed, but whether cases where the assumptions hold are insightful towards the behaviour of neural networks. We believe that we present a number of examples in this work across Section 3 on the XoR phase transition to the more complex contextual tasks which follow that show that our paradigm can lend such insight to settings which appear natural, and of broad interest to the field. Moreover, within the context of the prior linear dynamics work (Saxe et al. 2014, Saxe et al. 2019, Jarvis et al. 2023) it appears reasonable to think more insights can be found with our paradigm in future work. Importantly, we add no new assumptions to these prior works, which in most cases only considered linear network mappings. Yet these prior works, and other related works (Baldi & Hornik, 1989; Fukumizu, 1998; Arora et al., 2018; Lampinen & Ganguli, 2019) were able to clearly demonstrate the effect of adding a hidden layer on the learning dynamics of linear networks, for example. A unifying theory capable of handling all possible cases is indeed the goal, but on the way to such an achievement, the steps will be a number of different paradigms each with their own strengths and weaknesses. As we argue in the Introduction, the framework we present here can handle cases which violate the assumptions of all other theoretical paradigms present in the  literature. Thus, we believe this work makes a useful contributory step for theoreticians and can shed light on the behaviour of models for practitioners.

---

> ### Author Response · Authors · 2024-11-18
> **General Rebuttal by the Authors (Elaboration on Scalability)**
>
> To begin, we kindly direct the reviewers towards Lines 119 to 123, as well as Section 7 and Appendix B for more discussion on obtaining the gates for our model. Specifically, Lines 119 to 123 make clear that using a fixed gating strategy is motivated by the original GDLN framework (Saxe et al. 2022). Section 7 then clearly states that finding the gating structure is a limiting open problem but emphasises the utility of our paradigm as it is. Finally, Appendix B provides a simple algorithm for taking a step towards addressing this limitation, but also points to meta-learning as a future direction for addressing this limitation. Appendix B also notes the complexity which meta-learning brings and the possibility to learn gating patterns unavailable to a ReLU network. We argue that for these reasons, a proper treatment of meta-learning should be left to future work, placing it outside the scope of our present study. Furthermore, Section 6 demonstrates that we are sufficiently testing the paradigm and even shows an interpretable case where the paradigm is not able to fully explain the complexity of training a ReLU network. The tractability and control of our setting even make it clear that the addition of more depth is the cause of the networks difficult to completely learn the correlation-aligned features and the mechanism is the violation of Lemma 4.1. Thus, even in this case our model lends insight into the behaviour of nonlinear neural networks and points towards helpful directions of future work. Additionally, the tasks considered in Sections 4 to 6 are *significantly* more complex than any tasks previously considered in the deep linear network paradigm. In conclusion, we believe scaling to more complex settings appears premature, but we still provide a step forward for the deep linear network paradigm of theory and for theoretical approaches towards obtaining full training dynamics of neural networks in general.
>
> We conclude with a final point on the broader utility of our findings made with the ReLN framework - mixed selectivity and similar phenomena have been identified in multiple, much more naturalistic and large-scale settings  (Olah et al., 2017a; Lecomte et al., 2024; Locatello et al., 2019). As we mention in our discussion, mixed-selectivity is highly relevant (if not overlapping terminology) with polysemanticity (and monosemanticity) which is a concept that has garnered much attention recently with the massive LLM Claude (Templeton, et al. 2024). We draw this comparison to demonstrate that our findings are of interest to the broader machine learning community and not distinct from other large-scale experimental findings. We cite more works in both machine learning and neuroscience (which also has a long history discussing mixed-selectivity and its emergence in neural codes) which our work speaks to in the discussion (Anderson, 2010; Rigotti et al., 2013). In conclusion, while the tasks presented in this theoretical work necessarily remain tractable for analysis, in the context of the literature we believe our work provides a helpful contribution to a broader discussion of far more general applicability and impact to both machine learning and neuroscience.

---

### Author Response · Authors · 2024-11-28
**Revised Draft from Reviewer Feedback**

We thank the reviewers again for their feedback and helpful suggestions. An updated version of the manuscript has been uploaded where we *elaborate on the assumptions of our work* in a similar manner to the general comment here. We have commented on the assumptions in the main text and elaborated in Appendix A. In addition, in necessary places throughout the main text we have added some additional information to make clear the motivation for the tasks which we consider. Finally, we have included a table of hyper-parameters for the contextual tasks in the Appendix. All changes have been made in blue to make identifying and comparing these portions of the text easy.

We believe these small changes assist the clarity of our work greatly and address the main weaknesses of our work. Particularly, the useful elaboration on our assumptions and the fact that in many cases a useful derivation is possible with weaker assumptions. We thank the reviewers for their assistance on this matter.

---

### Meta-Review · Area_Chair_gzYz · 2024-12-20

**Metareview:**

This paper studies feature learning in finite-dimensional ReLU networks, primarily by establishing an equivalence with Gated Deep Linear Networks (GDLNs) and leveraging their tractability to derive learning dynamics. The analysis reveals finite-width ReLU networks' inductive bias towards structured mixed selectivity, whereby neural units exhibit activity across multiple contexts in structured ways.

The reviewers expressed concern with the strong assumptions regarding the data, model, and training trajectories, which may not hold in more complex scenarios or realistic datasets. Furthermore, the empirical validation is largely confined to simple, synthetic datasets, and the challenge in scaling to realistic datasets and architectures remains could limit the practical applicability and generalizability of the findings.

Despite these limitations, the work innovatively uses ReLNs to offer new insights into feature learning and structured mixed selectivity in ReLU networks, providing a theoretical framework that is supported by analytical solutions and empirical demonstrations on tailored tasks. This approach represents a novel contribution to the theoretical understanding of deep learning architectures, offering a mechanistic explanation for the emergence of structured, reusable latent representations. The community may find value in both the results and the methods, and therefore I recommend acceptance.

**Additional Comments On Reviewer Discussion:**

Reviewers criticized the study's reliance on restrictive assumptions, specifically regarding data structure and network alignment, and questioned the scalability of Rectified Linear Networks (ReLNs) to complex datasets beyond synthetic examples. They also sought clearer definitions of key terms like "feature learning" and raised concerns about the practicality of identifying gating structures in larger models. The authors responded by acknowledging limitations but emphasized that their assumptions are common in theoretical work and do not preclude useful analysis even when partially violated. They defended their framework's focus on feature learning, committed to clarifying definitions, and outlined future research directions including improved methods for gate identification and exploring the broader empirical validity of GDLNs, drawing parallels with the NTK framework to contextualize their contributions. They ultimately argued their work represents a valuable step towards understanding feature learning in finite ReLU networks despite current limitations.

While some reviewers may not be completely convinced by the rebuttal, there was general consensus that understanding feature learning is an important topic and that the current work provides a novel approach for attempting to do so. Even if the conclusions do not end up generalizing to more complex scenarios, the community will nevertheless benefit from learning about the novel approach and the interesting findings.

---

### Decision · Program_Chairs · 2025-01-22

Accept (Poster)